



# The IAS2024 coastal sea level dataset and first evaluations

Fukai Peng[1, *], Xiaoli Deng[2], Yunzhong Shen[3], Xiao Cheng[1]

[1] School of Geospatial Engineering and Science, Sun Yat-Sen University, Guangzhou 510275, China
[2] School of Engineering, The University of Newcastle, University Drive, Callaghan, NSW 2308, Australia
[3] College of Surveying and Geo-informatics, Tongji University, Shanghai 200092, China

*Correspondence to*: Fukai Peng (pengfk@mail.sysu.edu.cn)

**Abstract.** A new dedicated coastal sea level dataset called the International Altimetry Service 2024 (IAS2024, https://doi.org/10.5281/zenodo.13208305, Peng et al., 2024c) has been presented to monitor sea level changes along the world's coastlines. One of the reasons for generating this dataset is the quality of coastal altimeter data has been greatly improved with advanced coastal reprocessing strategies. In this study, the Seamless Combination of Multiple Retrackers (SCMR) strategy is adopted to obtain the reprocessed Jason data for January 2002 and April 2022. The evaluation results show that the data availability of the IAS2024 dataset over global coastal oceans is much higher than that of the European Space Agency Climate Change Initiative version 2.3 (ESA CCI v2.3) dataset. The closure of trend differences ($0.16\pm3.98$ mm yr$^{-1}$) between IAS2024 and Permanent Service for Mean Sea Level (PSMSL) tide gauge data is observed at the global scale, which further demonstrates the good performance of the IAS2024 dataset in monitoring the coastal sea level changes. The altimeter-based virtual stations have been built with the IAS2024 dataset over 0-10 km, 5-15 km and 10-20 km coastal strips, which will contribute to the analysis of coastal sea levels for the ocean community and the risk management for the policy makers. Our study also points out that no obvious variations exist in linear sea level trends from offshore to the coast in most coastal zones. In addition, the vertical land motion (VLM) estimates from the combination of coastal altimeter data with tide gauge records agree well with the University of La Rochelle 7a (ULR7a) Global Navigation Satellite System (GNSS) solution ($0.09\pm2.22$ mm yr$^{-1}$), suggesting that altimeter-derived VLM estimates can be an independent data source to validate the GNSS solutions.

## 1 Introduction

Satellite altimetry has become a mature remote sensing technique over open oceans since the launch of high-precision satellite altimetry missions (e.g., Topex/Poseidon). It has been making great contributions to quantifying and monitoring sea level changes at both global and regional scales (Ablain et al., 2016; Prandi et al., 2021; Guérou et al., 2023). Among these missions, the Jason altimetry series are usually used as the reference missions for monitoring global sea levels considering their high precision and stable performance. During the past two decades, the precision of reprocessed 20-Hz along-track altimeter data in coastal zones from Jason missions has been remarkably increased thanks to the dedicated coastal retrackers and improved range/geophysical corrections (Cipollini et al., 2017; Vignudelli et al., 2019; Peng et al., 2024a), which makes it possible to construct altimeter-based virtual stations along the world's coastlines (Benveniste et al., 2020; Cazenave et al., 2022).

The IAS (International Altimetry Service) Pilot Service was established in July 2023 as a service of the International Association of Geodesy (IAG) for providing information about altimetry data, geodetic and climate models, research and operational applications based on satellite altimetry technology innovations. One of its goals is to provide auxiliary data and algorithms to produce new products across the coastal zones including the coastal ocean, land-sea surface, estuaries and inland water bodies for applications in studies of the extremes, long-term climate change and environmental development.

In this paper, we present the IAS2024 coastal sea level dataset from Jason missions over the period of January 2002 and April 2022 reprocessed with the SCMR (Seamless Combination of Multiple Retrackers) strategy. A total of 1548 virtual stations are then built along the world's coastlines, which can be used to monitor the coastal sea levels and calculate the vertical land motion (VLM) estimates. In addition, the spatial variation of 20-Hz along-track sea level trends from offshore to nearshore is analysed using the IAS2024 dataset. The quality of reprocessed altimeter data is validated against the monthly tide gauge records from the PSMSL (Permanent Service for Mean Sea Level, https://psmsl.org/) and European Space Agency (ESA) Climate Change Initiative (CCI) coastal sea level dataset (version 2.3, https://www.seanoe.org/data/00631/74354/).

Section 2 presents the details of the SCMR processing strategy and methods to construct the altimeter-based virtual stations and to assess the altimeter data. Sections 3 and 4 illustrate the cross-validation results against both in-situ and independent altimeter dataset, as well as the application of coastal altimeter datasets. Sections 5 and 6 present the data availability and conclusions, respectively.

## 2 Computation of the IAS2024 coastal sea level dataset

### 2.1 The SCMR strategy

The IAS2024 coastal sea level dataset is generated using the SCMR (Peng et al., 2024b) strategy. The flow diagram is illustrated in Fig. 1, which starts from the waveform leading edge detection and ends with the seamless combination of sea surface height (SSH) estimates from multiple retrackers. The main steps are described in the following subsections.

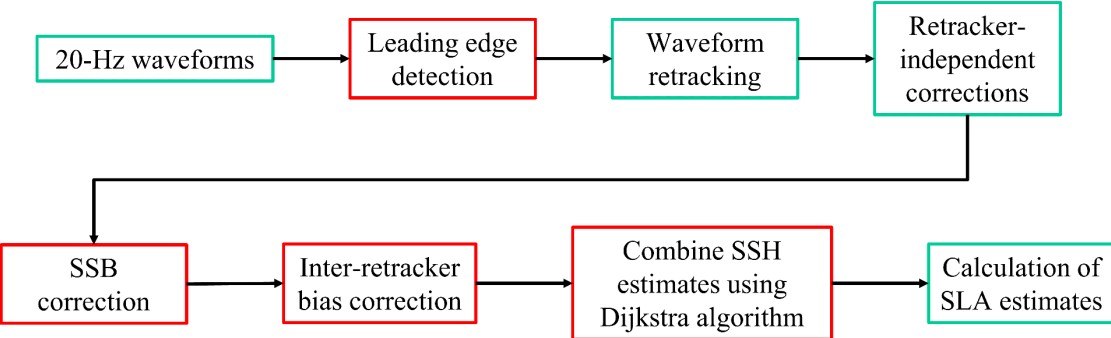

**Figure 1: Flow diagram of the SCMR processing strategy. The red boxes highlight the innovation points in the strategy.**





### 2.1.1 Detection of leading edges

The leading edge is a sequence of waveform gates, where the power increases sharply and thus can be detected based on the differences between consecutive power of gates. The start gate of the leading edge is deemed as the first gate where the power difference of two consecutive gates is positive (Passaro et al., 2014). Similarly, the stop gate of the leading edge is deemed as the first gate where the power difference of two consecutive gates is negative (Passaro et al., 2014) or smaller than a tiny positive number (Wang et al., 2019). Moreover, the power of the start gate should be lower than a constant level, and the power

difference between start and stop gates should be larger than a threshold value (Gommenginger et al., 2011). The main steps of the leading-edge detection method are as follows.

First, the intersection points between four horizontal lines with the power levels of 0.1, 0.2, 0.5 and 0.9 and the normalized waveform are derived. For each intersection point, the start and stop gates of the leading edge are searched forward and backward until satisfying the following equation,

$$P_{startgate} - P_{startgate-1} > 0.001 \quad\quad\quad (1)$$

$$P_{stopgate+1} - P_{stopgate} < 0.01$$

where the $P_{startgate}$ and $P_{stopgate}$ are the powers of the normalized waveform at the start and stop gates. If the difference (i.e., $P_{stopgate} - P_{startgate}$) is larger than 0.1 and the $P_{startgate}$ is lower than 0.2, the proportion of waveform from start to stop gates is selected as a potential leading edge. Here, the above threshold levels are selected based on previous studies and our

empirical experience (Passaro et al., 2014; Peng and Deng, 2020a).

Once all potential leading edges are found, the 50% Threshold retracker (Gommenginger et al., 2011) is applied to each potential leading edge to calculate the range and corresponding SSH estimates. The Dijkstra (1959) algorithm is applied to find the shortest path between offshore and nearshore along-track points, where the edge weights are the height differences between connected nodes (Roscher et al., 2017). As a result, the optimal SSH estimate for each 20-Hz along-track point and

the corresponding leading edge is determined (Peng et al., 2023). The advantage of this method is that it effectively avoids the perturbations before the true leading edge of the waveform (cf. Fig. 2 in Peng et al., 2024b).

### 2.1.2 Detection of land returns

When the altimeter approaches the coast, the contaminated waveforms appear with peaks that are caused by the high reflective areas inside the illuminated land surfaces or by the modification of the sea state close to the shoreline (Halimi et al., 2013).

This type of waveforms is denoted as Brown-peaky waveform, because the peaks caused by land returns are close to the waveform leading edge, while the waveform shape over oceans follows the Brown model (Fig. 4.5 in Gommenginger et al., 2011). Here, we detect the land returns following the idea of the adaptive peak detection (APD) method (Peng and Deng, 2018; Peng and Deng, 2020a) shown in Fig. 2, but with some modifications. The main steps are as follows.

Firstly, the forward moving average $F_t$ and backward moving average $B_t$ are derived using the normalized waveform,





$$F_t(i) = \frac{1}{N}\sum_{j=0}^{N-1} P_t(i+j), \quad i = 1, 2, \cdots, n - N + 1 \tag{2}$$

$$B_t(i) = \frac{1}{N}\sum_{j=0}^{N-1} P_t(i-j), \quad i = N,\ N+1, \cdots, n$$

where the $N$ is the moving average step, which is selected as five through our trial-and-error test, $n$ is the number of waveform gates, which is 104 for Jason missions.

Next, we define the Difference I as $(F_t - B_t)$. This is first used to determine the Leftfoot points, which correspond to the local

maxima of Difference I whose height and prominence are larger than 0.12 using the MATLAB function "findpeaks". This is then used to determine the Leftedge, Rightfoot and Rightedge points. The Leftedge and Rightedge points are the zero-crossing points of Difference I before and after the corresponding Leftfoot point. The Rightfoot points are the local minima between consecutive Leftfoot points (Fig. 2). If there exist multiple Leftfoot points, the waveform is denoted as the multi-peak waveform (Figs. 2a and 2b). Otherwise, the waveform is classified as the Brown-peaky waveform if the Difference I of the

Rightfoot point is smaller than −0.2 (Figs. 2c and 2d).

Finally, the start gate of land returns is defined as the stop gate of the leading edge, while the stop gate of land returns is the last zero-crossing point between Rightfoot and Rightedge points (black line in Fig. 2d). Note that the modified APD method used here is dedicated for the Brown-peaky waveforms instead of the multi-peak waveforms.

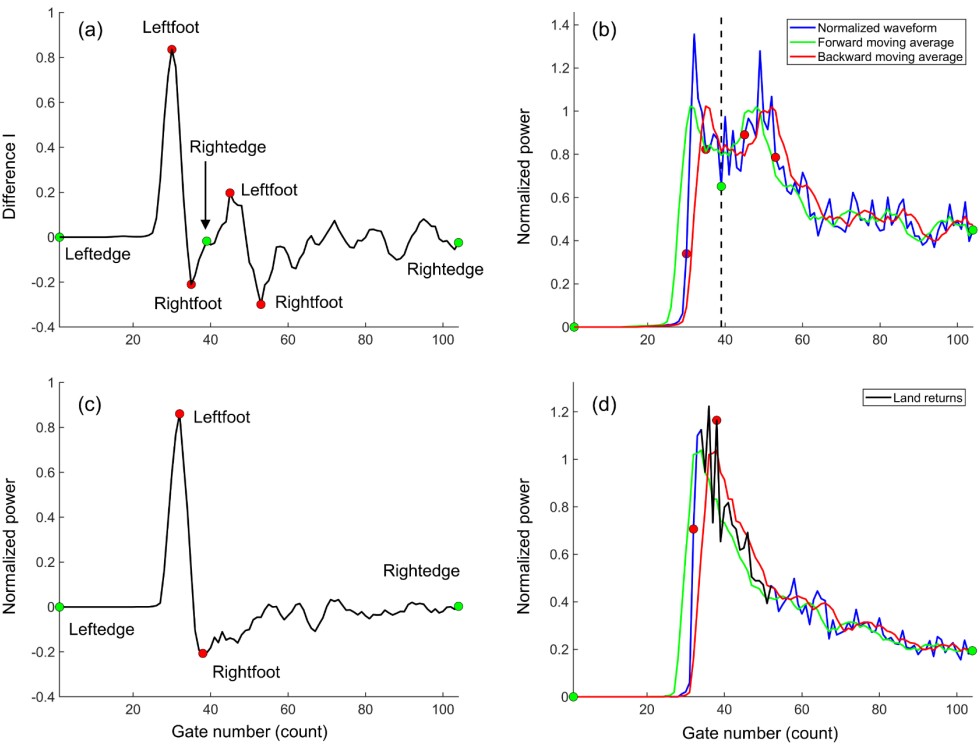

**Figure 2: Illustration of the adaptive peak detection (APD) method. The subplots (a) and (b) show the curve of Difference I and feature points detected by the APD method for multi-peak waveforms, respectively. The subplots (c) and (d) show the similar results for Brown-peaky waveforms. The Difference I is calculated as forward moving average of the normalized waveform minus backward moving average of the normalized waveform. The black line corresponds to the gates affected by the land returns.**

### 2.1.3 Waveform retracking

Four different retrackers are used in this study, which are the official Sensor Geophysical Data Record (SGDR) Maximum Likelihood Estimator 4-parameter (MLE4) retracker, the Adaptive Leading Edge Sub-waveform (ALES, Passaro et al., 2014), the Weighted Least Squares 3-parameter (WLS3, Peng and Deng, 2018) and the adaptive retracker (Poisson et al., 2018). All of the retrackers can process ocean waveforms. In addition, the ALES retracker can handle waveforms with land returns observed in the waveform trailing edge, the WLS3 retracker can cope with the Brown-peaky waveforms with the downsized weights (i.e., 0.01) being assigned to land returns determined by the modified APD method, and the adaptive retracker can deal with the specular waveforms (Fig. 3). Note that the estimates from the SGDR MLE4 are directly provided by the official product, while the estimates from the ALES, WLS3 and adaptive retrackers are solved independently.

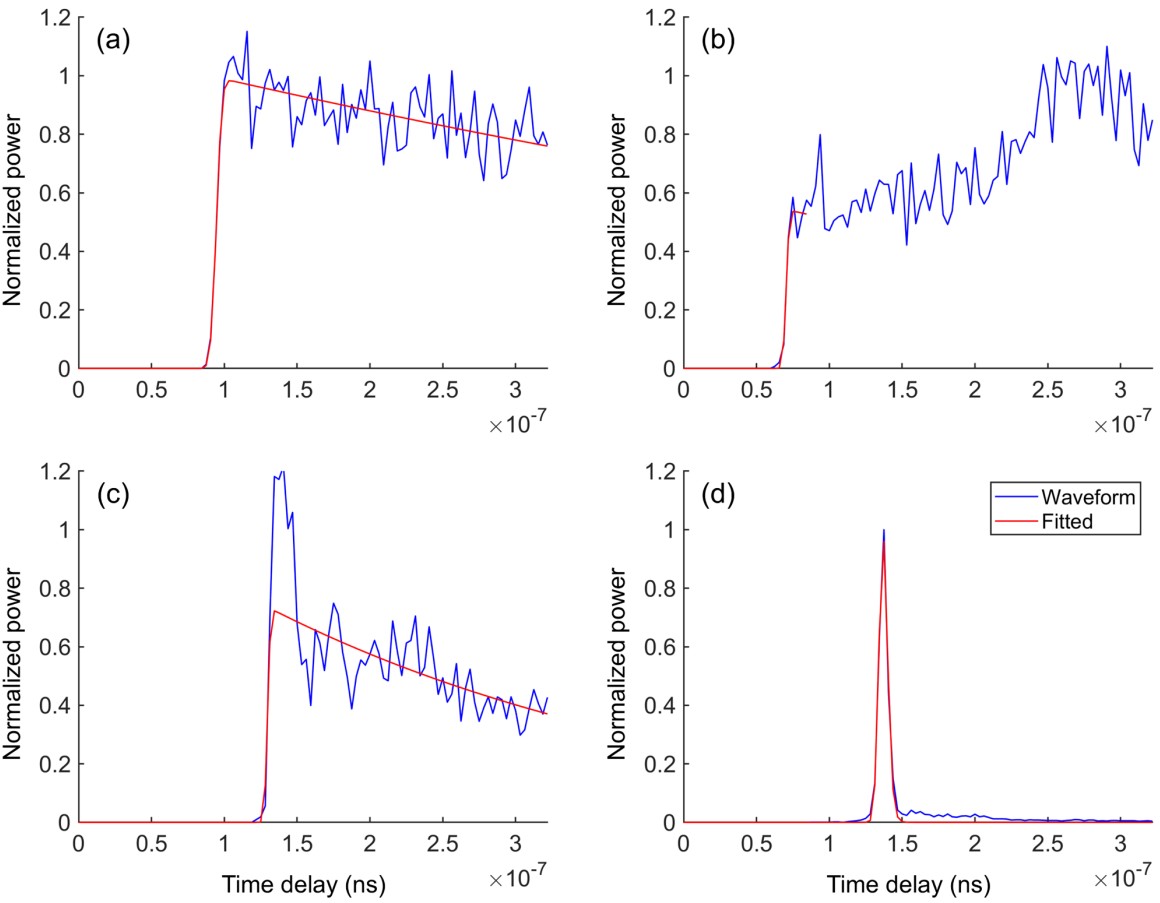

**Figure 3: An example of observed waveforms and fitted waveforms from different retrackers. The subplots (a)-(d) show the results for different waveform types by methods of (a) SGDR MLE4, (b) ALES, (c) WLS3 and (d) adaptive retrackers.**



### 2.1.4 Calculation of altimeter SSH

The altimeter SSH at each 20-Hz along-track point is calculated as follows,

$$SSH = h_{alt} - R_r - R_{cor} \tag{3}$$

where the $h_{alt}$ is the height of the satellite above the Topex ellipsoid, the $R_r$ is the distance between the altimeter and nadir sea surface derived from the waveform retracker, the $R_{cor}$ include dry and wet tropospheric, ionospheric, and sea state bias corrections. The appropriate corrections for each altimetry mission are investigated and summarized in Table 1.

**Table 1. Range/geophysical corrections applied to Jason missions used in this study.**

| Corrections | Jason-1/2/3 |
|---|---|
| Dry tropospheric correction | ECMWF |
| Wet tropospheric correction | GPD+ (Fernandes et al., 2016) |
| Ionospheric correction | GIM (Komjathy and Born, 1999) |
| Sea state bias | Peng and Deng, (2020b) |
| Geocentric ocean and loading tide | FES2014 (Lyard et al., 2021) |
| Dynamic atmospheric correction | MOG2D (Carrère and Lyard, 2003) |
| Solid earth tide | Cartwright and Taylor (1971) |
| Pole tide | Desai et al., (2015) |
| Mean sea surface | CLS22 |

### 2.1.5 Seamless combination of SSH estimates from multiple retrackers

The seamless combination of SSH estimates from multiple retrackers consists of two steps. First, the biases of SSH estimates with respect to the official SGDR MLE4 are calculated and removed using an inter-retracker bias correction model (Peng et 130 al., 2021). Second, the Dijkstra (1959) algorithm is applied to the networks constructed by the height differences between connected nodes to obtain the optimal along-track SSH profile based on the assumption that the altimeter along-track SSH estimates do not change significantly at the spatial scale of ~300 m (Cipollini et al., 2017). The inter-retracker bias correction model is as follows (Peng et al., 2024b),

$$\Delta h = \rho \times \Delta H_s + c_b \tag{4}$$
$$h_{cor} = \hat{\rho} \times \Delta H_s + \hat{c_b} \tag{5}$$



where $\Delta H_s$ and $\Delta h$ are the SWH differences and SSH differences with respect to the SGDR MLE4 retracker (e.g., ALES minus SGDR MLE4), $\rho$ and $c_b$ are the linear regression slope and intercept parameters, which are unknown and need to be estimated. Once estimates of two parameters, $\hat{\rho}$ and $\hat{c_b}$, are determined, the inter-retracker bias correction, $h_{cor}$, can be calculated by substituting SWH differences $\Delta H_s$ into the Equation (5). After removing the inter-retracker bias, the most appropriate SSH profile between the offshore point and the point closest to the coast is derived by using the Dijkstra algorithm, in which the edge weight is defined as the absolute SSH difference between two connected nodes.

**2.1.6 Calculation of altimeter SLA**

The altimeter SLA estimate at each 20-Hz along-track point is calculated as follows,
$$SLA = SSH - G_{cor} - MSS \tag{6}$$
where the $G_{cor}$ include geocentric ocean and loading tide corrections, dynamic atmospheric correction, solid earth tide and pole tide corrections, the $MSS$ is the interpolated value from the CLS22 mean sea surface (MSS) model using the bilinear interpolation.

**2.2 Assessment of the IAS2024 coastal sea level dataset**

The assessment of the IAS2024 coastal sea level dataset is conducted as follows. Firstly, the data availability and precision of 20-Hz SLA estimates are calculated as a function of distance to the coast. The data availability is calculated as the percentage of available 20-Hz SLA estimates, while the data precision is represented as the median value of standard deviation (STD) of 20-Hz SLA estimates (Cipollini et al., 2017).

Secondly, the power spectrums of 20-Hz SLA estimates are computed with the periodogram method (Zanifé et al., 2003). The SLA power spectrum reflects the strength of SLA signals at different spatial scales and can be used to estimate the noise level from the high frequency part of the spectrum. The crossover analysis (Gaspar et al., 1994) is also conducted to examine the spatial variation of data quality over global coastal oceans. For each single mission, the crossover point is the intersection of two crossing ground tracks. The 20-Hz along-track points closest to the crossover point are used to derive the collocated SLA estimates at the crossover point using the linear interpolation method. To reduce the effect of temporal variability, the collocated SLA estimates are only considered if their time lags are within one day.

Finally, the monthly SLA time series from the IAS2024 dataset are compared with those from the ESA CCI v2.3 dataset over global coastal oceans. In addition, both of them are validated against monthly tide gauge records from PSMSL. The main steps are described as follows.

1) A single nominal ground track is divided into multiple ground track segments if the distance between consecutive 20-Hz along-track points is larger than 10 km. Only the ground track segments whose nearshore points with a distance to the coast smaller than 10 km and offshore points with a distance to the coast larger than 20 km are used in this study. The 20-Hz along-track SLA estimates from the IAS2024 dataset are referenced to the corresponding ground track segments, and a 3-sigma filter is then applied to the along-track SLA estimates to remove the outliers.





2) The inter-mission SLA biases between different Jason missions are estimated using the overlapping time series method (Waston et al., 2011) and removed to construct an SLA time series with a temporal sampling of 10 days. A 3-sigma filter is applied to the SLA time series and only the SLA time series whose percentages of available data are higher than 80% are used to derive the monthly SLA time series over the period of January 2002 and April 2022.

3) The 20-Hz along-track SLA estimates from the ESA CCI v2.3 dataset are also referenced to the corresponding ground track segments before generating the monthly SLA time series over the period of January 2002 and June 2021.

4) The linear sea level trends are derived from the deseasoned monthly SLA time series with the Hector software (Bos et al., 2014), which can handle the time series with temporally correlated noise. The stochastic noise models used in this study include the first- and fifth-order autoregressive (i.e., AR1 and AR5), and the autoregressive fractionally-integrated moving-average (ARFIMA) models. The most appropriate noise model is identified using the lowest mean value of both the Akaike Information Criterion (AIC; Akaike, 1992) and the Bayesian Information Criterion (BIC; Schwarz, 1978).

5) The correlation coefficients, root mean square (RMS) of the differences between deseasoned and detrended monthly SLA time series from the IAS2024 and ESA CCI v2.3 datasets are calculated. The trend differences between these two datasets, as well as trend uncertainties of these two datasets are also used for comparison. Although the time spans for the IAS2024 and ESA CCI v2.3 datasets are slightly different, it has little impact on the linear sea level trend estimates from our test.

6) The monthly SLA time series of 20-Hz along-track points over the 10-20 km coastal strip are averaged to generate mean monthly SLA time series at altimeter-based virtual stations. The collocated virtual station and tide gauge are derived when their distance is smaller than 200 km and the score between them reaches the maximum. The score is calculated using the correlation coefficient and RMS of the differences between the monthly SLA time series from the virtual station and tide gauge as follows,

$$score = score_{cc}^i \times 0.4 + score_{rms}^i \times 0.6 \tag{7}$$

$$score_{cc}^i = \frac{cc_i - min(cc)}{max(cc) - min(cc)} \times 100$$

$$score_{rms}^i = \frac{max(rms) - rms_i}{max(rms) - min(rms)} \times 100$$

The number of virtual stations, the number of collocated virtual stations and tide gauges, the correlation coefficients and RMS of the differences between deseasoned and detrended monthly SLA time series from them, and the differences of linear sea level trends derived from the deseasoned monthly SLA time series between them are used for the validation.



### 2.3 Application of the IAS2024 coastal sea level dataset

Three applications of coastal altimeter datasets are presented in this study. Firstly, the coastal altimeter datasets can be used to build altimeter-based virtual stations (Benveniste et al., 2020; Cazenave et al., 2022), which is of great importance for monitoring the coastal sea level change and constraining the high-resolution ocean models. To examine the data quality of virtual stations obtained from different coastal strips, the monthly SLA time series of 20-Hz along-track points over 0-10 km, 5-15 km and 10-20 km are averaged for comparison. For clarity, these three types of virtual stations are denoted as onshore, nearshore and offshore virtual stations.

Secondly, the spatial variation of sea level trends can be analysed because the altimeter provides the along-track sea level trend profile, which would help us understand the relative contributions of local and remote processes on coastal sea levels. To address this issue, a linear regression is applied to the along-track sea level trends using the MATLAB function "robustfit". The trend differences between nearshore and offshore points are then calculated to examine whether there exists significant change ($>\pm2$ mm yr$^{-1}$) of the along-track sea level trends.

Finally, the VLM estimates can be obtained by calculating the trend estimates from differenced monthly SLA time series between collocated virtual stations and tide gauges, which is used for the comparison with corresponding Global Navigation Satellite System (GNSS) solutions called the University of La Rochelle 7a (ULR7a) (Gravelle et al., 2023). The ULR7a solution is a preliminary version of the reanalysis of 21 years of GPS data from 2000 to 2020 that has been undertaken within the framework of the third data reprocessing campaign of the International GNSS Service (IGS), whose associated vertical velocity field is expressed in the International Terrestrial Reference Frame 2014 (ITRF 2014).

### 3 Evaluation

#### 3.1 Data availability and precision

Figure 4 shows the data availability and precision for the official SGDR MLE4 data and SCMR-reprocessed data over different Jason missions. These results are derived after removing the SLA estimates whose absolute values are greater than 1 m. As we can see from the graph, the results from different Jason missions are similar, demonstrating the stability of the Jason missions. Thanks to the Open Loop tracking mode (Biancamaria et al., 2018), the Jason-3 can recover more reliable SLA estimates than the other two satellites. The results also demonstrate that the SCMR strategy outperforms the official SGDR MLE4 over global coastal oceans. The SCMR can remarkably increase the data availability over the entire 0-20 km distance bands, as well as the data precision beyond 6 km to the coast. Note that the higher precision achieved by the official SGDR MLE4 within 0-6 km distance band can be attributed to the low data availability (Figs. 4a to 4c) and dedicated data editing strategy applied to the official SGDR product (Peng et al., 2024b). The improved percentage in terms of data availability and precision in different coastal strips (i.e., 0-5 km, 5-10 km and 10-20 km) are summarized in Table 2.

As can be seen from Table 2, the improved percentage for data availability increases with decreasing offshore distance. In contrast, the data precision shows a declining trend from offshore oceans towards the coast. This result demonstrates that the SCMR strategy can handle multiple waveform types and thus can recover more reliable SLA estimates than the official SGDR
MLE4. Within distance 0-5 km to the coast, the data precision with respect to the official SGDR MLE4 is not improved, which is mainly attributed to the above reasons. In addition, the range estimates from contaminated waveforms and range/geophysical corrections near the coast are inferior to those over open oceans, which results in a smaller or even negative improvement percentage of data precision. Therefore, a dedicated data editing strategy should be developed for the coastal altimeter data as what has been done in the SGDR product. Overall, the data availability from SCMR can retain more than 90% beyond 5 km
offshore and more than 70%-80% in the last 5 km to the coast (Figs. 4a-4c). The data precision can retain at centimetre levels (~4.9 cm) over the 10-20 km distance band, increase to 7-9 cm over the 5-10 km distance band and rise up to decimetre levels (~20-23 cm) towards the coast (Figs. 4d-4f).

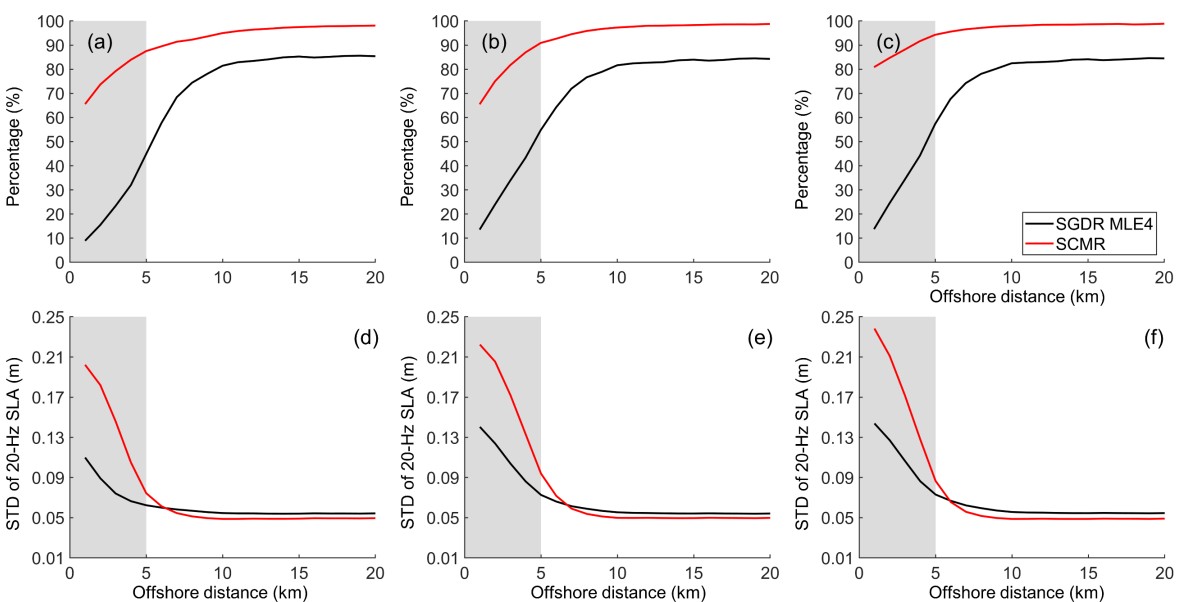

**Figure 4: Data availability (a-c) and precision (d-f) for official SGDR MLE4 data (in black) and SCMR-reprocessed data (in red). From left to right, the results for Jason-1, Jason-2 and Jason-3 are shown successively. The data precision is represented as the median value of the standard deviation of 20-Hz SLA estimates within one second in each 1-km distance band.**

**Table 2. Improved percentage of the SCMR against the official SGDR MLE4 in terms of data availability and precision over different**
**coastal strips.**

| Satellite altimeters | 0-5 km | | 5-10 km | | 10-20 km | |
|---|---|---|---|---|---|---|
| | Availability | Precision | Availability | Precision | Availability | Precision |





| | | | | | | |
|---|---|---|---|---|---|---|
| Jason-1 | 53.0% | −72.4% | 20.3% | 7.0% | 12.6% | 9.2% |
| Jason-2 | 46.1% | −54.7% | 20.7% | 4.8% | 14.7% | 8.4% |
| Jason-3 | 53.2% | −51.9% | 20.4% | 10.1% | 14.7% | 10.5% |

The power spectrum and crossover analysis (Fig. 5) further demonstrate the good performance of the SCMR strategy in improving the data availability and reducing noise levels at small scales. Compared to the official SGDR MLE4, the SCMR achieves a reduction in the SLA spectra at the scales below 50 km, with the noise level at sub-kilometre scales being 23% less than that of the official SGDR MLE4 (Figs. 5a-5c). In addition, the crossover analysis shows that the SCMR significantly increases the number of collocated 20-Hz SLA estimates by 44%, while retaining smaller mean differences close to zero and slightly larger standard deviations around 14 cm (Figs. 5d-5f).

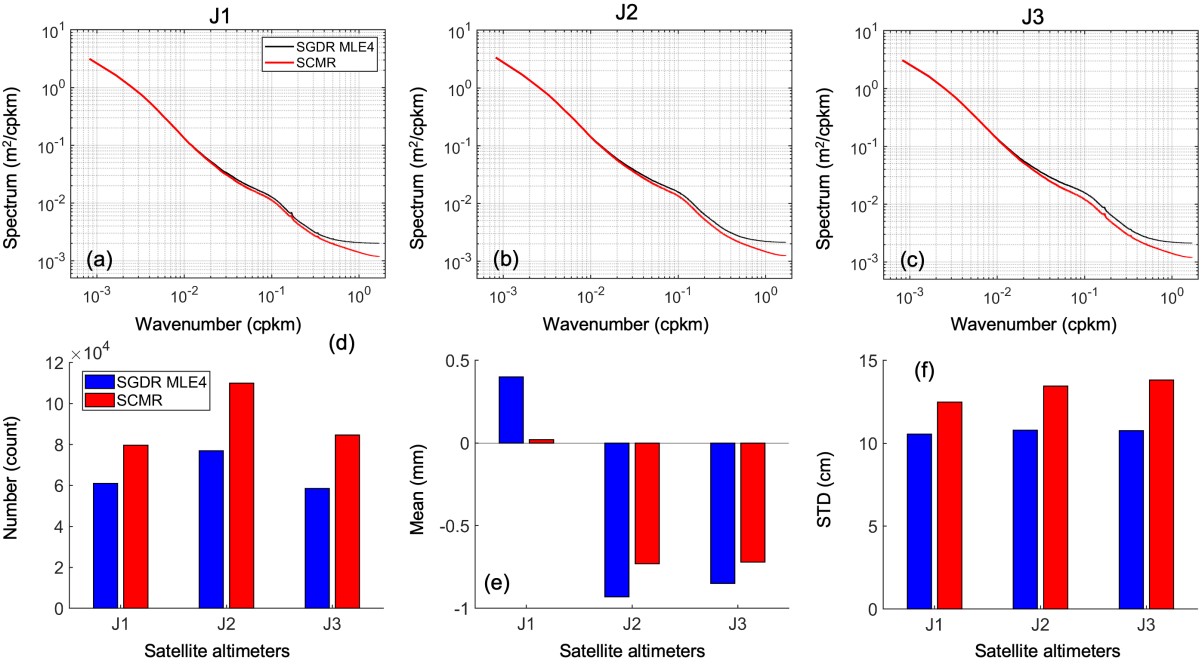

**Figure 5: Power spectrum of 20-Hz SLA estimates (a-c) and number, mean and standard deviation of differences from collocated 20-Hz SLA estimates at crossover points (d-f). J1, J2 and J3 represent Jason-1, Jason-2 and Jason-3 missions, respectively.**

## 3.2 Comparison between IAS2024 and ESA CCI v2.3

In this section, the quality of the IAS2024 coastal dataset is investigated by comparing the latest ESA CCI v2.3 coastal dataset. To achieve this goal, we first calculate the point-wise correlation coefficient and RMS of the differences between monthly SLA time series from these two datasets and obtain the averaged value for each ground track segment. Then, the point-wise


along-track sea level trends from these two datasets over a similar period are also derived and the trend differences are averaged for each ground track segment. Finally, the point-wise monthly time series within the 10-20 km coastal strip of a single ground track segment are used to generate the mean monthly time series at altimeter-based virtual stations for both datasets. The mean monthly time series are validated against the tide gauge records in terms of correlation coefficients, RMS values and trend

differences.

Figure 6 shows the histogram of correlation coefficients and RMS values between detrended and deseasoned monthly SLA time series between IAS2024 and ESA CCI v2.3 datasets for different ground track segments over global coastal oceans. As we can see from the graph, the consistency of the monthly SLA time series between these two datasets is not as good as expected at the global scale, which contrasts with our previous study in the Australian coastal zone (Peng et al., 2022).

Although the relatively high correlation coefficients (>0.4) are achieved in most cases, there still exist low (either positive or negative) correlation coefficients. As a result, the mean correlation coefficient between IAS2024 and ESA CCI v2.3 datasets is only 0.44, while the corresponding RMS values mostly varying between the range of 60 mm and 80 mm. The spatial distribution of both correlation and coefficients and RMS values (Fig. 7) reveals that the discrepancy between IAS2024 and ESA CCI v2.3 datasets is mainly observed in the mid-to-high latitudes (>40°N and <40°S) and islands over open oceans.

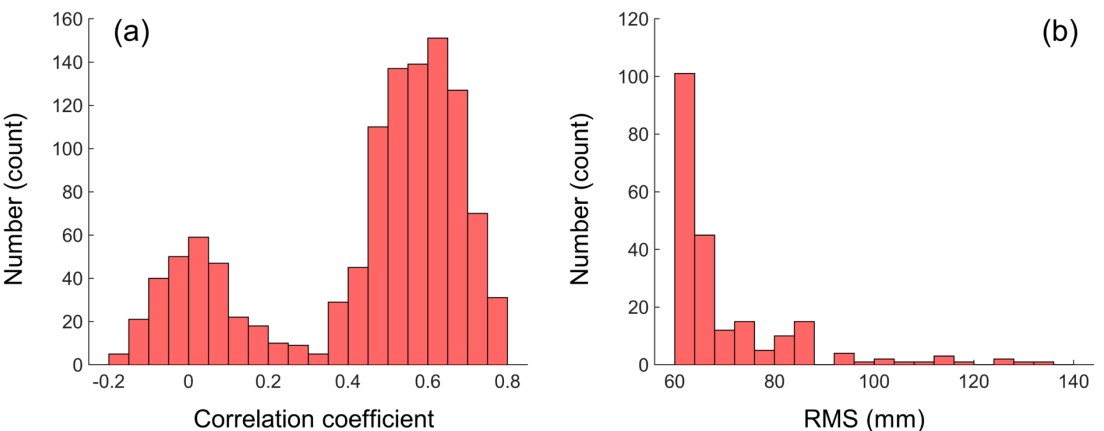


**Figure 6: Distribution of correlation coefficients (a) and RMS values (b) between detrended and de-seasoned monthly SLA time series from IAS2024 and ESA CCI v2.3 datasets for different ground track segments over global coastal oceans.**

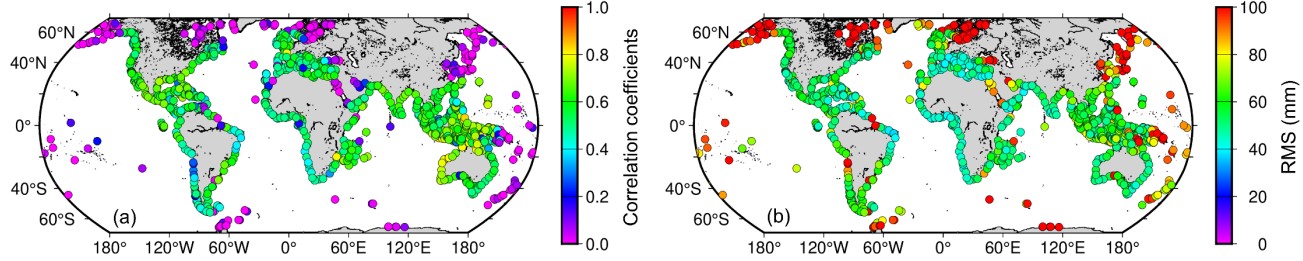

**Figure 7: Spatial distribution of correlation coefficients (a) and RMS values (b) between detrended and de-seasoned monthly SLA**
**time series from IAS2024 and ESA CCI v2.3 datasets for different ground track segments over global coastal oceans.**



Further investigation shows that the trend differences between them are predominantly within the range between −4 mm yr⁻¹ and 6 mm yr⁻¹, with a mean of trend differences being 1.32±2.39 mm yr⁻¹ (Fig. 8a). The trend uncertainties for both datasets are similar, varying between 0.5 mm yr⁻¹ and 2.0 mm yr⁻¹ (Fig. 8b). This result implicates that the sea level trends derived from the IAS2024 dataset are higher than those from the ESA CCI v2.3 dataset, which is in line with our previous findings (Peng et al., 2022; Peng et al., 2024a). The reason behind this may be attributed to the different data processing techniques adopted, especially the methods used to estimate the inter-mission biases (Peng et al., 2022). To explore which dataset is more preferable, the tide gauge records from the PSMSL are deemed as the ground truth and used for the validation of these two datasets. Table 3 summarizes the validation results in term of the number of virtual stations, the number of collocated virtual stations and tide gauges, mean correlation coefficients and RMS values, and the mean of trend differences between altimeter and tide gauge.

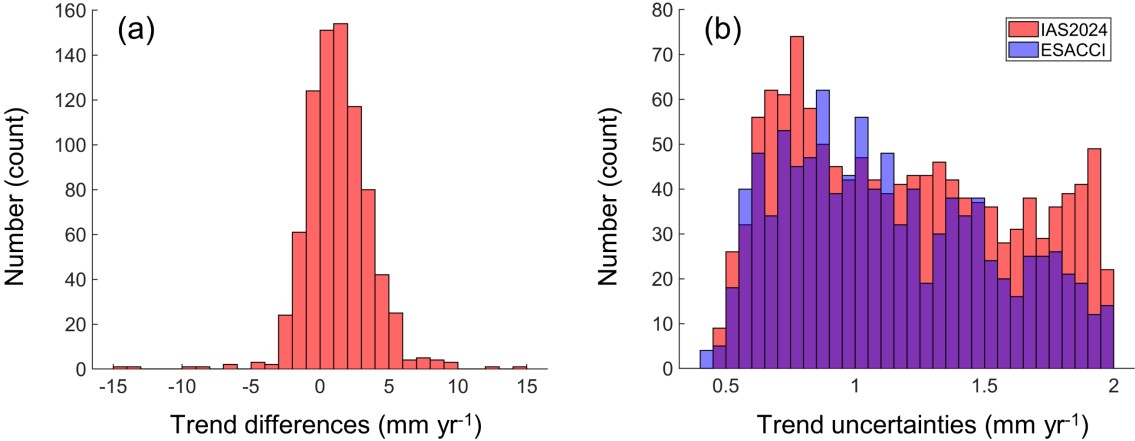

**Figure 8: Distribution of (a) trend differences from deseasoned monthly SLA time series between IAS2024 and ESA CCI v2.3 datasets, as well as (b) the trend uncertainties for different ground track segments over global coastal oceans.**

**Table 3. Validation results of IAS2024 and ESA CCI v2.3 datasets against tide gauge records from PSMSL. Note that the correlation coefficients and RMS values are derived from the detrended and deseasoned monthly SLA time series, while the sea level trends are derived from the deseasoned monthly SLA time series.**

| Altimeter datasets | Number of virtual stations | Number of collocated stations | Correlation coefficient with tide gauges | RMS (mm) | Trend difference (mm yr⁻¹) |
|---|---|---|---|---|---|
| IAS2024 | 1548 | 497 | 0.67 | 43 | 0.17±3.91 |
| ESA CCI v2.3 | 705 | 152 | 0.61 | 37 | −1.67±3.42 |

As illustrated by the table, the number of virtual stations, as well as the number of collocated stations, for the IAS2024 dataset (1548 and 501) are much higher than those for the ESA CCI v2.3 dataset (705 and 152). This is because the IAS2024 dataset achieves more 20-Hz data points over the global coastal oceans than those by the ESA CCI v2.3 product, and thus the former have more collocated stations for validation against nearby tide gauge stations. Despite the number of collocated stations is





different, the similar performance is observed for these two datasets in terms of correlation coefficients and RMS values. However, these two datasets differ in the trend differences between altimeter and tide gauge at the global scale. As reported by previous studies (e.g., Waston et al., 2015; Wöppelmann and Marcos, 2016), the VLM would be cancelled out on average

along the world's coastlines. Therefore, it is reasonable to assume that the average trend difference between the altimeter and tide gauge is close to zero at the global scale. The result from the IAS2024 dataset is thus consistent with this assumption, as the mean of trend differences is 0.17±3.91 mm yr$^{-1}$, which demonstrates the good performance of the IAS2024 dataset in monitoring the coastal sea levels.

However, a negative trend difference (−1.67±3.42 mm yr$^{-1}$) between the ESA CCI v2.3 dataset and tide gauges is found in

Table 3. This may be ascribed to the small number of collocated stations (152) with tide gauges used for the estimation. This would give rise to a bias because the above assumption is unlikely to be held at regional and local scales (Wöppelmann and Marcos, 2016). Considering that the sea level trends from the ESA CCI v2.3 are 1.32±2.39 mm yr$^{-1}$ lower than those from the IAS2024 dataset on average, and the trend difference for the ESA CCI v2.3 would become a more reasonable value (i.e., −0.35±4.17 mm yr$^{-1}$) if the trend bias is corrected. In addition, when taking only 152 collocated virtual stations and tide gauges

for the IAS2024 dataset, the trend difference becomes −0.63±3.66 mm yr$^{-1}$, which is similar to the bias-corrected value of the ESACCI v2.3 (i.e., −0.35±4.17 mm yr$^{-1}$). Therefore, we do think that the sea level trends from the ESA CCI v2.3 are slightly underestimated. This finding is coherent with the results revealed by two regional cases in the Australian coastal zone (Peng et al., 2022) and the northern South China Sea (Peng et al., 2024a).

## 4 Applications of coastal altimeter datasets

The coastal altimeter datasets are an important complement to the existing tide gauge networks considering their higher spatial coverage (Fig. 9). As we can see from the graph, the altimeter-based virtual stations are distributed along the world's coastlines, while the tide gauges are mostly situated in Japan, Australia, North America and Western Europe. Moreover, the altimeter measures the absolute sea levels relative to the ITRF, which would not be affected by the VLM and thus would help us investigate the spatial variation of sea level changes from the offshore oceans to the coast. In this section, we focus on three

applications of coastal altimeter datasets. The first is to build altimeter-based virtual stations along the world's coastlines. The second is to investigate whether there are significant trend differences between offshore and nearshore. The former is of paramount importance for coastal management and risk adaptation (Fox-Kemper et al., 2021), while the second would give us insights into the role of small-scale processes in changing the coastal sea level trends (Woodworth et al., 2019; Harvey et al., 2021; Cazenave et al., 2022). The last is to calculate the VLM estimates by combining virtual stations and tide gauges, which

provides a way to validate the VLM estimates from GNSS stations (Wöppelmann and Marcos, 2016).

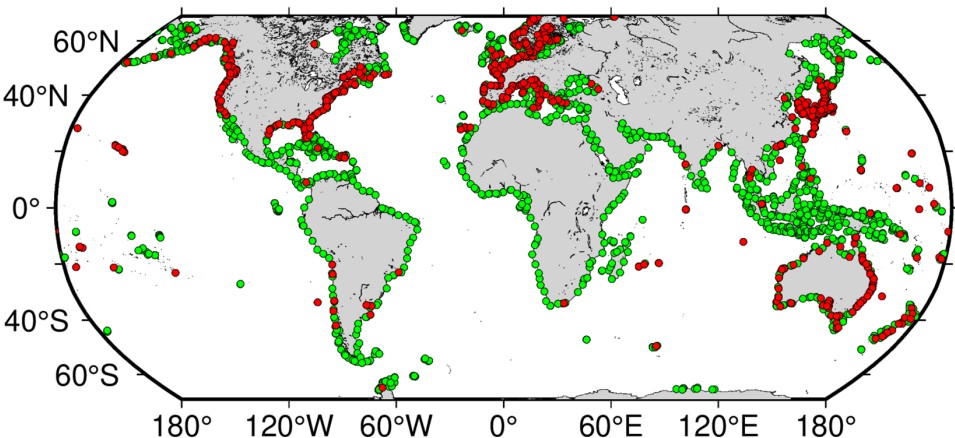

**Figure 9: Distribution of the IAS2024 altimeter-based virtual stations (green solid circles) from Jason missions and tide gauges (red solid circles) from PSMSL.**

## 4.1 Altimeter-based virtual stations

To better monitor the coastal sea levels, it is expected that the virtual stations are located as close to the coast as possible. However, the precision of altimeter data degrades with decreasing offshore distance to the coast (Fig. 4). Therefore, it is necessary to investigate which coastal strip is more preferable for building the virtual stations. Here, we generate three different sets of virtual stations following the methods described in Section 2.2 using the IAS2024 dataset, and refer them as the onshore, nearshore and offshore virtual stations according to their distances to the coast. Note that the onshore, nearshore and offshore stations are generated from altimeter data within 0-10 km, 5-15 km and 10-20 km distance bands, respectively. The ESA CCI v2.3 dataset is not used here considering their small number of data points over the global coastal oceans.

Figure 10 shows the comparison results between onshore virtual stations and tide gauges in terms of correlation coefficients and RMS values. As can be seen from the graph, the high correlation coefficients (>0.6) and low RMS values (<60 mm) are observed along the world's coastlines, which indicates the good quality of the onshore virtual stations. The slightly better results are achieved by the nearshore and offshore virtual stations (Table 4). These results suggest that the monthly sea level signals are highly correlated over the spatial scales of ~20 kilometres. Therefore, the correlation coefficient between nearshore and offshore time series can be an important indicator to evaluate the data quality of nearshore. It is also noted that the low correlation coefficients and high RMS values are observed at some onshore stations such as the east of North America and the west of South America (Fig. 9), which can be attributed to three reasons. Firstly, the VLM would affect the comparison between the tide gauge and altimeter at the local scale (Wöppelmann and Marcos, 2016). Secondly, the collocated virtual stations and tide gauges are usually tens to hundreds of kilometres away, from which they would measure different sea level signals (Vinogradov and Ponte, 2010). Finally, the coastal environments (e.g., bathymetry, sea states and morphology) can be quite different from those offshore regions, which leads to the significant deterioration of range estimates and range/geophysical

corrections (e.g., geocentric ocean and loading tide corrections, wet tropospheric correction, sea state bias correction and dynamic atmospheric correction) (Gommenginger et al., 2011; Abele et al., 2023; Peng et al., 2024b).

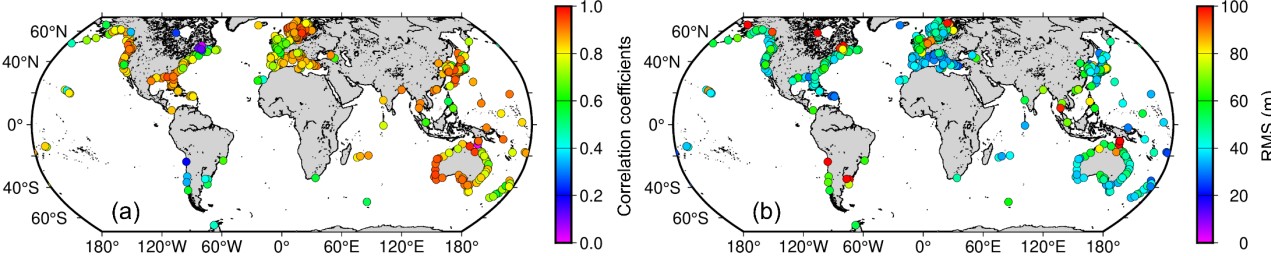

**Figure 10: Comparison results in terms of (a) correlation coefficient and (b) root mean square (RMS) of the differences between**
**monthly time series from onshore virtual stations and tide gauges. The onshore virtual stations are generated using 20-Hz along-track altimeter data within 0-10 km coastal strip.**

Compared to the nearshore and offshore stations, the onshore stations are more likely affected by land contamination because they include the altimeter data within 5 km of the coast, leading to the smaller number of virtual stations (Table 4). However,
the observation degradation seems not to be remarkable and the closure of the trend difference between satellite and tide gauge at the global scale is unchanged. This result implies that the altimeter data within 0-5 km coastal strip can be used when the dedicated editing strategy is taken. In addition, the sea level trends obtained from altimeter data closer to the coastline are more consistent with those from tide gauge records at the global scale. Therefore, we can draw a conclusion that the 0-10 km coastal strip is the most suitable for building the virtual stations because of the trade-off between the closest distances to the coast and
the reliability of the data quality.

**Table 4. Comparison results between altimeter-based virtual stations and tide gauges. The correlation coefficients and RMS values are derived from the monthly SLA time series between virtual stations and tide gauges, while the trend estimates are obtained from the deseasoned SLA time series.**

| Statistics | Onshore | Nearshore | Offshore |
|---|---|---|---|
| Mean distance to the coast | 6.5 km | 8.8 km | 13.4 km |
| Number of virtual stations | 1534 | 1548 | 1548 |
| Number of collocated stations | 497 | 497 | 497 |
| Correlation coefficient | 0.80 | 0.81 | 0.82 |
| RMS (mm) | 51.71 | 49.13 | 47.15 |
| Trend difference (mm yr$^{-1}$) | 0.16±3.98 | 0.16±3.96 | 0.17±3.92 |



## 4.2 Spatial variations of sea level trends towards the coast

As reported by previous studies (Deng et al., 2011; Benveniste et al., 2020; Harvey et al., 2021), the coastal sea level trends measured by tide gauges differ from altimeter-derived sea level trends offshore. However, it is uncertain whether the trend discrepancy is due to the different ocean dynamic processes measured by the tide gauge and altimeter or the VLM (Cazenave et al., 2022). Therefore, a good way to solve this issue is to explore the spatial variation of altimeter-derived sea level trends from offshore to nearshore. For example, Cazenave et al. (2022) compared the coastal trends (within 4-5 km) with the offshore trends (within15-17 km) and found no significant difference (within $\pm 2.0$ mm yr$^{-1}$) at 78% of the 756 virtual stations. Note that the degradation of altimeter data within 5 km of the coast would affect the results from the above method. To avoid this problem, a robust linear regression is applied to the along-track trends using the MATLAB function "robustfit" in this study. This regression function is chosen because it can alleviate the impact of outliers on the estimation procedure from our experimental tests. The trend differences between nearshore and offshore points are then calculated. If the absolute trend differences are smaller than 2.0 mm yr$^{-1}$, it is considered that there is no significant discrepancy between nearshore and offshore sea level trends. Otherwise, the increasing (decreasing) trend is detected if the trend difference is larger than 2.0 mm yr$^{-1}$ (smaller than $-2.0$ mm yr$^{-1}$).

Figure 11 presents four examples of 20-Hz along-track sea level trends against distance to the coast for both IAS2024 and ESA CCI v2.3 datasets. As we can see from the graph, the spatial variations of 20-Hz along-track trends show different patterns (i.e., constant, increasing or decreasing) depending on different geolocations. In addition, the smooth variations of 20-Hz along-track trends at the spatial scales of ~300 m is observed (Figs. 11a to 11c). Therefore, the abrupt fluctuation of 20-Hz along-track trends near the coast (Fig. 11d) can be attributed to the degradation of altimeter data.

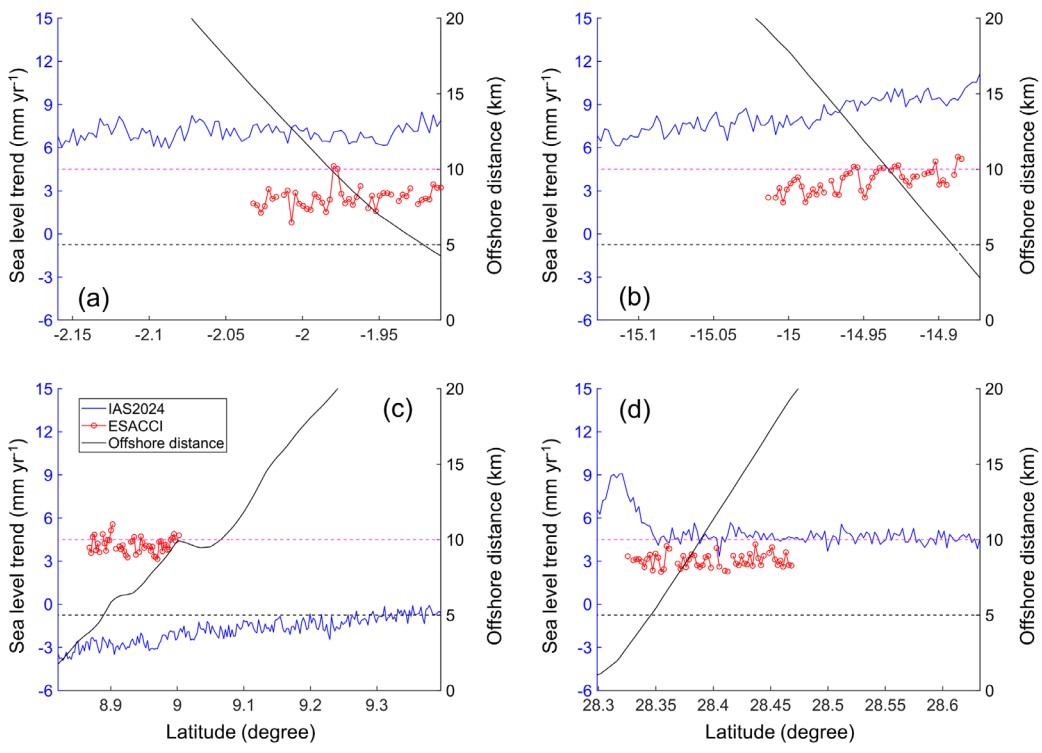

**Figure 11: Examples of 20-Hz along-track sea level trends against distance to the coast for both IAS2024 and ESA CCI v2.3 datasets. The subplots (a) to (d) show constant, increasing and decreasing trends, as well as abrupt fluctuation towards the coast. Horizontal dash lines indicate distances to the coast at 5 km (in black) and 10 km (in magenta).**

The statistical results of trend differences over global coastal oceans for these two datasets are illustrated in Fig. 12. As demonstrated by the graph, there is no significant difference (within ±2 mm yr$^{-1}$, i.e., of the maximum level of trend uncertainties) at 95% of the 1548 ground track segments. In the remaining 5%, we observe either an increasing trend (2%) or a decreasing trend (3%). This result is consistent with that from the ESA CCI v2.3 dataset, where 82% of the 807 ground track segments show no trend difference, and the percentage of decreasing trend (9%) is equal to that of increasing trend (9%). It is also observed that there is no any concentration of coastal trends departing from offshore trends in a particular region (Fig. 13), which means that the increasing or decreasing trends can occur along the world's coastlines.

Earth System
Open Access   Science
Data   Discussions

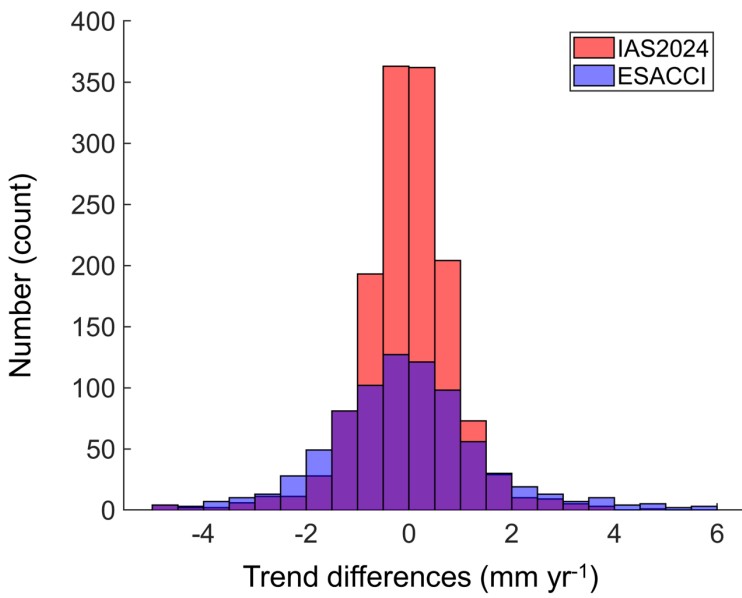

**Figure 12: Trend differences between nearshore and offshore for the IAS2024 and ESA CCI v2.3 datasets.**


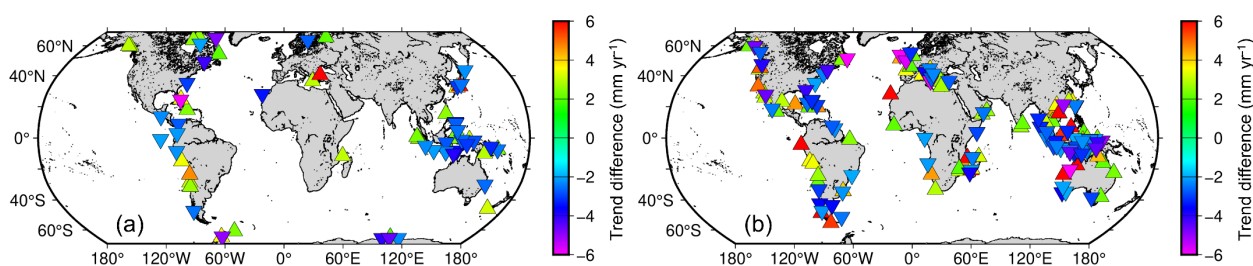

**Figure 13: Trend differences between nearshore and offshore for the IAS2024 (a) and ESA CCI v2.3 (b) datasets. Triangles and inverse triangles correspond to increasing and decreasing trends, respectively.**

The trend differences in most places are insignificant, which may be ascribed to two aspects. Firstly, most of the trend uncertainties are relatively large (~0.5-2.0 mm yr$^{-1}$), which is insensitive to the small changes of sea level trends. Secondly, the small-scale processes (e.g., coastal currents, winds, waves, freshwater input in river estuaries) induced variations do not significantly affect the coastal sea level trends, because the sea level variation over longer time scales is more likely affected by the remote signals from open oceans (Han et al., 2019). It is also noted that there still exist some places where the trend

differences are remarkable, which motivates us to investigate the reasons behind the trend variation. For instance, a recent study by Piecuch et al., (2018a) has demonstrated the role of river discharge in affecting the coastal sea levels over interannual or longer time scales. Moreover, the high-resolution ocean models (grid mesh smaller than 1 km) would contribute to the systematic quantification of coastal phenomena, causing the reported trend to increase or decrease near the coast (Marti et al.,



2019; Gouzenes et al., 2020; Dieng et al., 2021). However, high-resolution ocean models are currently not available over
global coastal oceans (Han et al., 2019; Melet et al., 2020; Laignel et al., 2023). Therefore, more efforts should be made in the
future to improve our understanding of present-day coastal sea level changes and to improve the ability of climate models to
simulate future sea levels in highly-populated and vulnerable coastal regions of the world.

### 4.3 Vertical land motion

The combination of satellite radar altimetry with tide gauge data can be used to estimate the VLM (e.g., Wöppelmann and
Marcos, 2016). Figure 14 shows the sea level trends at virtual stations and tide gauges, respectively. Note that the result for
the ESA CCI v2.3 dataset is not shown due to the small number of virtual stations (see Section 3.2). As we can see from the
graph, the results for different sets of virtual stations are consistent along the world's coastlines except in some sporadic points
along the south coast of Greenland and the coast of Antarctica. The contamination from sea ice and ice sheets may be the
causes that underlie the observed sea level trend discrepancies (Davis, 1995). It is also noted that the remarkable differences
between the altimeter and tide gauge are concentrated in the coasts of the Gulf of Mexico and Northern Europe. This is because
Northern Europe suffers from large uplift rates due to the Glacial Isostatic Adjustment (GIA), while the northern Gulf of
Mexico is exposed to large coastal subsidence rates due to the subsurface fluid withdrawal (Wöppelmann and Marcos, 2016;
Piecuch et al., 2018b; Harvey et al., 2021).


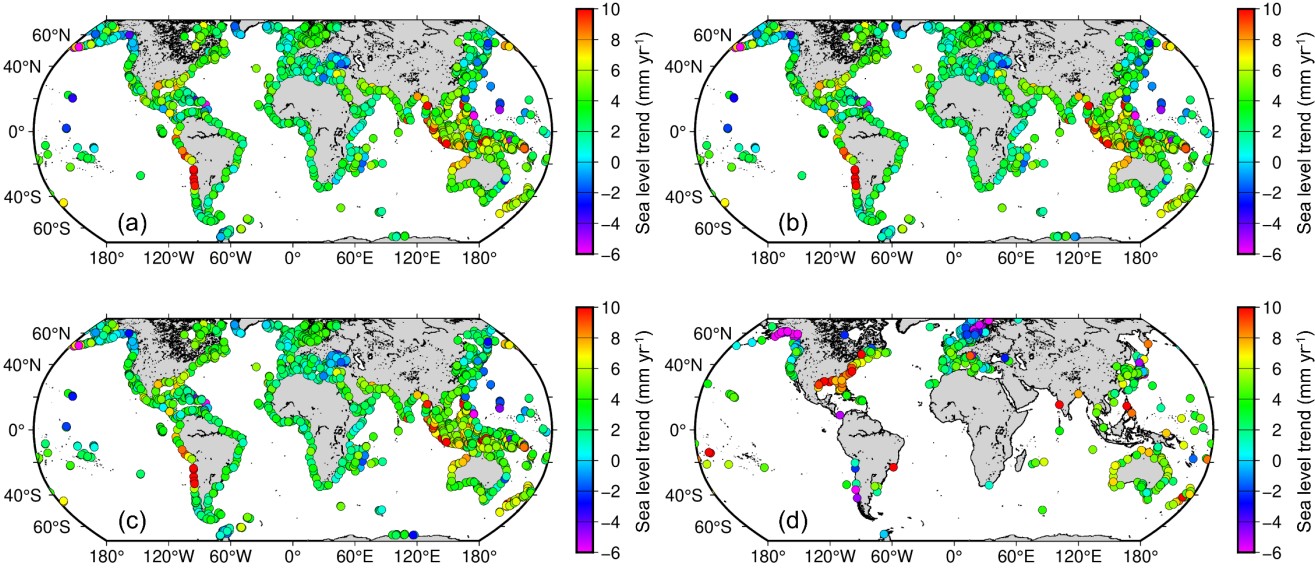

**Figure 14: Sea level trends over the period of January 2002 and April 2022 at altimeter-based virtual stations and tide gauges along the world's coastlines. The subplots (a) to (c) show the results of onshore (0-10 km), nearshore (5-15 km) and offshore (10-20) virtual stations from the IAS2024 dataset, while the subplot (d) presents the results from the PSMSL tide gauges.**



The VLM derived from the difference between the deseasoned monthly SLA time series of the altimeter and tide gauge (i.e., ALT−TG) can better illustrate this (Fig. 15). As can be seen, the large uplift rates (>2 mm yr$^{-1}$) are observed in the Northern Europe and northwest coast of Canada, suggesting that the GIA-related radial crust displacement are significant in these regions. In addition, the upper lift rates are also observed along the southwest coast of South America. The large subsidence rates (<−2

mm yr$^{-1}$), however, are presented along the coast of the northern Gulf of Mexico and Southeast Asia where the groundwater extraction is significant (Wöppelmann and Marcos, 2016; Harvey et al., 2021). Along the Australian coastline, both the uplift and subsidence rates are observed but dominated by subsidence rates along the northeast coast. The results of onshore stations are similar to those of nearshore and offshore stations, demonstrating the role of small-scale processes in changing the coastal sea levels is not significant. When it comes to the Western Europe and Japan, the VLM estimates show large variations

dependent on different locations, with the overall median values being close to zero for different sets of virtual stations.

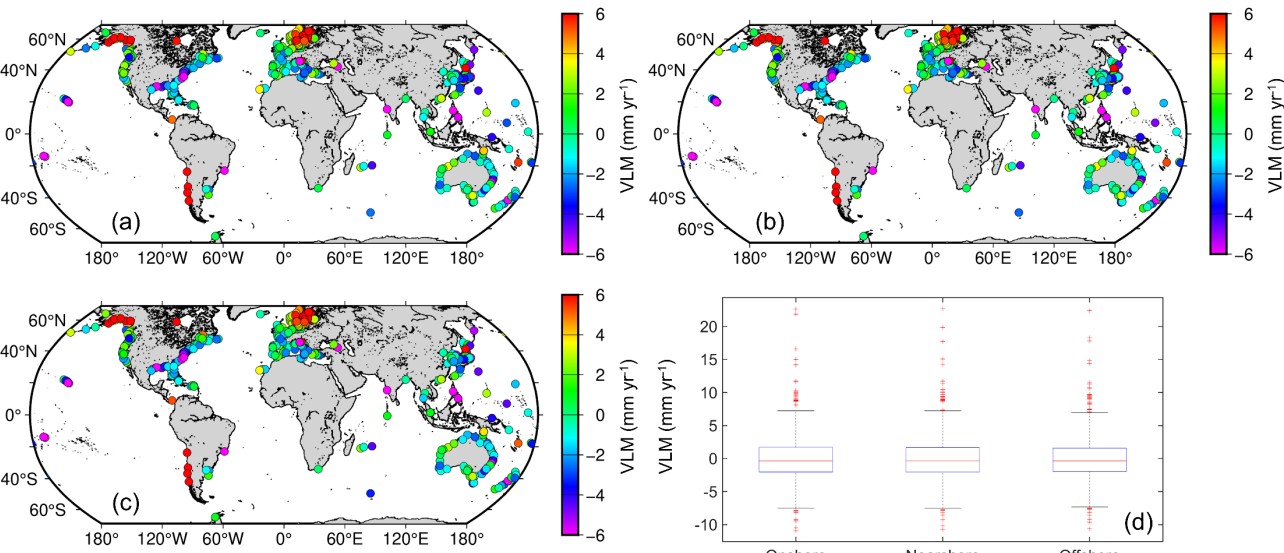

**Figure 15: VLM estimates derived from the combination of virtual stations and tide gauge. The subplots (a) to (c) show the results for onshore, nearshore and offshore virtual stations, respectively. The subplot (d) shows the boxplot of the VLM estimates.**


Figure 15d illustrates that the VLM estimates from onshore virtual stations are in accordance with those from nearshore and offshore stations with median values being close to zero. This result further demonstrates that coastal and offshore trends are identical in most cases, which is consistent with the results shown in Section 4.2. To assess the robustness of the results, the VLM estimates from GNSS observations are used for comparison (Fig. 16). As we can see from the graph, the VLM estimates

from virtual stations agree well with those from GNSS observations in most cases with the differences being smaller than ±1.5 mm yr$^{-1}$. The mean of the differences is 0.09 mm yr$^{-1}$ with an STD of 2.22 mm yr$^{-1}$ for onshore virtual stations, while the similar values are 0.13 (0.14) mm yr$^{-1}$ and 2.17 (2.13) mm yr$^{-1}$ for nearshore (offshore) virtual stations. Therefore, the results




from altimeter-based virtual stations can be used as an independent source to validate the VLM estimates from GNSS observations.


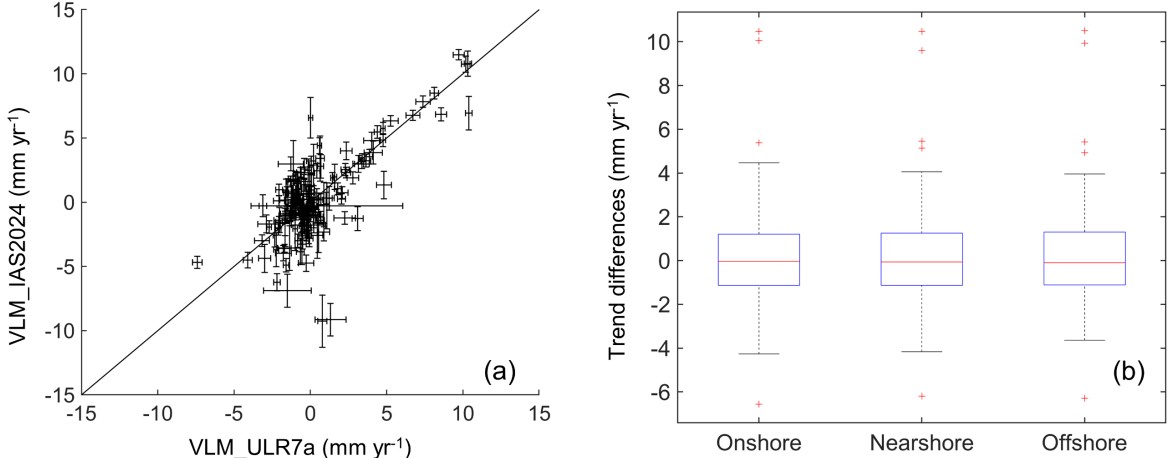

**Figure 16: Comparison between ULR7a-derived VLM estimates from GNSS observations and IAS2024-based virtual stations. The subplot (a) shows the scatterplot between IAS2024 onshore virtual stations and GNSS stations, while the subplot (b) shows the corresponding boxplot.**

**5 Data availability**

The IAS2024 coastal sea level dataset is available for open access at https://doi.org/10.5281/zenodo.13208305 as network Common Data Form (netCDF) files (Peng et al., 2024c). This dataset includes monthly SLA time series at 20-Hz along-track points for different ground track segments and at onshore, nearshore and offshore virtual stations. Moreover, the corresponding SLA time series at 10-day cycle time scales. The FES2014 geocentric ocean and loading tide corrections, the MOG2D dynamic

atmospheric correction, and the CLS22 MSS over the same time scales are also provided for studying oceanography and geodesy.

**6 Conclusions**

A new reprocessed coastal sea level dataset, namely IAS2024, for monitoring sea level changes along the world's coastlines has been presented, along with the first evaluations. The Seamless Combination of Multiple Retrackers (SCMR) processing

strategy is adopted to generate the coastal 20-Hz SLA estimates from Jason missions over the period of January 2002 and April 2022. The main conclusions of this study are summarized as follows.

The SCMR strategy has significantly increased the data availability when compared to the official SGDR MLE4 retracker, with the improvement percentage varying between 12.6% and 53.2% depending on different coastal strips. The moderate





improvement (4.8%-10.1%) of data precision brought by the SCMR strategy is also observed, mainly because the SCMR can

reduce the variability of the SLA spectrum below the wavelength of 50 km. As a result, the data availability of SCMR-reprocessed Jason data can retain more than 90% beyond 5 km offshore and more than 70%-80% onshore within 5 km to the coast. In addition, the data (20-Hz) precision can be retained at centimeter levels (5-9 cm) over the 5-20 km distance band and at decimetre levels (~20-23 cm) towards the coastline. Therefore, the IAS2024 dataset generated with the SCMR strategy has the potential to provide reliable SLA estimates for monitoring the coastal sea level changes.

The performance of the IAS2024 dataset has been evaluated through comparisons of the monthly SLA time series, as well as sea level trend estimates, with those from the ESA CCI v2.3 dataset and the PSMSL tide gauge records. The inter-comparison between IAS2024 and ESA CCI v2.3 datasets shows that only a moderate correlation coefficient (0.44) is achieved at the global scale, with the RMS values mostly ranging from 60 mm to 80 mm. The large discrepancy is mainly observed in mid-to-high (<40°S and >40°N) latitudes and islands over open oceans. The sea level trends from the IAS2024 dataset are on

average $1.32\pm2.39$ mm yr$^{-1}$ higher than those from the ESA CCI v2.3. These may be attributed to the different data processing techniques adopted, especially the methods used to estimate the inter-mission biases. The validation against tide gauges demonstrates that the IAS2024 dataset not only retrieves much more sea level data than the ESA CCI v2.3 dataset over global coastal oceans but also achieves the closure of trend differences ($0.16\pm3.98$ mm yr$^{-1}$) at the global scale, very close to the theoretical value of zero. This result demonstrates the good performance of the IAS2024 dataset in monitoring the coastal sea

levels. In contrast, a negative trend difference ($-1.67\pm3.42$ mm yr$^{-1}$) is found for the ESA CCI v2.3, which may be due to the small number of sampling points or the underestimated sea level trends. Therefore, the performance of the ESA CCI v2.3 dataset cannot be comprehensively evaluated in this study.

Three applications of the IAS2024 dataset are then conducted in this study, which would give us insights into how dedicated coastal sea level datasets can be used for ocean communities and policymakers. Firstly, the altimeter-based virtual stations are

built along the world's coastline over 0-10 km, 5-15 km, 10-20 km coastal strips, which are denoted as onshore, nearshore and offshore virtual stations, respectively. The results show that virtual stations achieve much higher spatial coverage than the PSMSL tide gauges, which would contribute to the analysis of coastal sea level changes in places where tide gauge data are unavailable. It is also found that the onshore stations would be affected by the degradation of altimeter data within 5 km of the coast. Nevertheless, the impact is not significant and the onshore stations can be thus invaluable for monitoring the global

coastal sea levels.

Secondly, the spatial variations of linear sea level trends are evaluated with both the IAS2024 and ESA CCI v2.3 datasets. The results from these two datasets show a similar pattern but with different magnitudes at the global scale, with the constant trend being dominant (95% for IAS2024 and 82% for ESA CCI v2.3). This result indicates that the local processes have little impact on coastal sea level changes over longer time scales, which is consistent with the findings revealed by Cazenave et al., (2022).

Besides, the reasons leading to the increasing or decreasing trends found in some places should be further investigated with the help of high-resolution ocean models, which would be our future work. Finally, the VLM estimates derived from the combination of altimeter and tide gauge are consistent with those from the GNSS stations (e.g., $0.09\pm2.22$ mm yr$^{-1}$ for onshore

stations). Therefore, the altimeter-derived VLM estimates can be used as an independent data source for validating the GNSS solutions.

**Author contributions**

FKP conducted conceptualization, investigation, data curation, developed methodology to process the altimeter data using MATLAB software, validated and visualized the results, and funding acquisition. XLD helped with the formal analysis of the methods and results. YZS and XC provided the resources. All authors contributed to the manuscript writing.

**Competing interests**

The contact author has declared that none of the authors has any competing interests.

**Disclaimer**

Publisher's note: Copernicus Publications remains neutral with regard to jurisdictional claims in published maps and institutional affiliations.

**Acknowledgments**

The authors are thankful to the space agencies for providing the high-quality altimeter data from Jason missions to users.

**Financial support**

This research is supported partially by the National Natural Science Foundation of China (Grant No. 42106175), the Natural Science Foundation of Guangdong Province (Grant No. 2022A1515011299), and the Fundamental Research Funds for the Central Universities (Grant No. 02502150056). This research is also supported partially by the Australian Research Council's
Discovery Projects funding scheme (Project No. DP220102969).



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
