# Peer review of "The IAS2024 coastal sea level dataset and first evaluations"

_Earth System Science Data, 2024_

## Referee Comment (RC2)

**Report on 'the IAS2024 coastal sea level data set and first evaluations' by Fukai Peng et al.**

This study proposes a new reprocessing of altimetry data from Jason-1, 2 and 3 in the world coastal zones over 2002-2022. It provides a new useful data set of altimetry-based sea level time series at a large number of coastal sites, complementing the limited tide gauge network. The coastal data are provided along satellite tracks at 20hz resolution, from the coast up to 20 km offshore. They are based on retracking of the radar waveforms using 3 different retrackers (ALES, WLS3, Adaptative Retracker), in addition to the standard retracker MLE4 used for the open ocean geophysical data records (GDRs). A total of 1548 'virtual' coastal stations are built along the world coastlines. The data set is validated by using available tide gauge data. It is also compared to a previously published, similar coastal product (called here ESA CCI product).

In principle, this proposed new dataset merits to be published. However, important clarification concerning the data processing is needed. The comparison with the ESA CCI product also requires significant clarification. Thus, the manuscript requires major revision before being suitable for publication in ESSD. Overall, the manuscript lacks a lot of crucial information about the data processing. Besides, the recurrent claim, all along the paper, that the IAS2024 is superior than the ESA CCI product (which is possibly true but not fully convincing) suggests competing interests rather than the desire of objectively presenting a new useful product.

**Major and minor concerns are listed below (with major concerns in bold).**

- Page 6, Calculation of altimeter SSH: To my knowledge the **GPD+ wet tropospheric correction** is not available beyond mid-2021. Which correction is applied beyond that date?
- Page 6, Seamless combination of SSH estimates from multiple retrackers: it is not clear **how the different retrackers are used**. What is the bias (wrt the MLE4-based GDRs) you are talking about? Is it estimated for each 20Hz measurement? What is the Dijkstra algorithm? In fine, **how are the ranges estimated? Are they based on different retrackers depending on the shape of the radar waveforms?** Please clarify. I think it would be useful to provide some discussion on the respective performances of the retrackers. The retracking procedure is crucial for building coastal altimetry data sets.
- Page 7, Assessment of the IAS2024 dataset : After your 3-sigma editing, how are the **temporal distribution of the data per mission**? For example, the Jason-1 mission suffers more for missing data than Jason-2 and Jason-3. **These missing data affect trend estimates**. Please discuss.
- Page 8, Assessment of the IAS2024 dataset: Little is said about the **intermission bias estimation**. Since you are focusing on coastal altimetry data, how do you compute it? Locally? regionally? This point is also crucial because any error in the intermission bias directly affects SLA trend estimates.
- Page 8 and beyond, Assessment of the IAS2024 dataset : **How is the coast defined**? Which data set is used and **how is the distance to the coast computed**?

  Do you apply a smoothing to your SLA time series?

  Could you show the distribution of closest distance to the coast for the 0-10km segments?

- Page 9 and beyond , Application of the IAS2024 data set: You define three sets of along-track segments (0-10km, 5-15 km, 10k-20 km) and assign virtual stations to each segment. **From Table 4 (page 16), I note about the same number of virtual stations are found for each segment, which is surprising.** I would expect more virtual stations for the offshore segment (10-20 km) than for the onshore segment (0-10km) because more noisy data should have been eliminated in the latter case. Please discuss.

- There is no indication on how the 10-day data are projected onto fixed points along a nominal ground track. How do you proceed?
- Page 10, Figure 4: indicate on the panels the name of the satellite. The figure caption should be more detailed.

- **Page 11-14: Comparison with the ESA CCI data set: I have a number of comments on this comparison.**
    1. First of all, it is not clear at all which version of the ESA CCI product has been used here. Page 8, it is written that the ESA CCI v2.3 data set is used (with data provided over January 2002 to June 2021). If this is correct, why only ~800 sites are correlated in Figure 6.a (I exclude the left hand part of the histogram centered at zero which seems to correlate data from IAS2024 at sites where ESA CCI has no data; i.e., in high latitude regions, Red Sea, tropical islands –see my comment below-). The ESA CCI v2.3 version contains 1189 virtual stations. Thus the histogram should show 1189 counts. The authors should indicate the date at which they downloaded the ESA CCI data to be clear about which product they considered.
    2. It is not fully clear how the ESA CCI data are used for the comparison. From what is written page 12, lines 261-264, it looks like that the ESA CCI along-track SLAs are averaged over the same 3 segments as for the IAS2024 data and then compared. But according to Fig.6a 'correlation coefficient plots' this seems not be the case. How many common points are considered in each segment? Does this comparison refer to the common time span of the two data sets?
    3. Fig.6a is totally unclear and likely misleading. To which virtual stations correspond the two histograms? As the processing of the raw coastal altimetry data are rather similar (except that IAS2024 uses several retrackers compared to ESA CCI which only uses ALES, but all other corrections being the same), **I do not understand the large number of cases with zero correlation. This seems totally unlikely**. My understanding is that the left hand side histogram of Fig.6a centered at zero corresponds to the purples dots of Figure 7 (i.e., where the ESA CCI product has no data). If this is the case, then the histogram centered at zero is not only misleading but definitely wrong! Please remove it!
    4. Please show examples to SLA time series at the same location from the two products. If, for example, your SLA time series are smoothed and if you compared to unsmoothed ESA CCI data, the correlation will indeed decrease. Please explain exactly how you have done.
    5. **Fig.7a is misleading**. As the ESA CCI product has no data in high latitude regions, red sea and tropical islands, **the map should show the correlation ONLY at the common sites**. All purple points (where ESA CCI has no data) need to be removed to not give the false impression that there are numerous sites where the correlation is zero.
    6. Fig.7b: **same comment as for Fig.7a. It is misleading to show high rms (red points) where the ESA CCI product has no data!**
    7. Page 13: Comparison with ESA CCI; **trend differences**. As for the SLA time series, **it is totally unclear how the comparison is done**. To trust this comparison exercise, you must be clear on what you do.
    I recommend you **provide a comparison of your trends computed over the 10-20km segments with trends estimated from classical gridded products** (e.g., Copernicus datasets, CMEMS or C3S). This would be another highly useful way of validating your product (in terms of trends). This has been done for the ESA CCI product for the offshore data (>10km from the coast) with trend differences < 1-2 mm/yr in general with the C3S gridded product.

- Page 14-16, Altmeter-based virtual stations: comparison with tide gauges. **How many tide gauge are used**? Justify the **200 km distance threshold** considered for the virtual stations-tide gauge comparison. Do the tide gauge records have continuous data over the overlapping time span or not? If not, how do you proceed? Please explain.
- Table 4 as well as Fig.14d and page 21: Concerning the trend comparison, what means 'trend differences'? Do the authors compare VLMs estimated from the differences altimetry-based SLAs and tide gauges, and GNSS-based VLMS? If this is the case, how many tide gauges have collocated GNSS stations? In many cases, GNSS-based VLM solutions often show significant discrepancies between computing centers. You chose the ULR7a solution here, which is fine. But do you have an idea of the associated VLM uncertainties?
- Pages 20-21, Vertical land motions: This section estimates VLMs from the differences altimetry-based SLAs and tide gauge data. This is ok. But over the rather short time span of analysis (20 years), **many coastal SLAs display strong interannual variability, masking any long term trend**. Only a few coastal regions show dominant linear sea level trends over this time span. Thus what are the errors associated with the trend estimates by this approach?
- The plots in Figures 15d and 16b are unclear. What are supposed to show the boxes?
- The color bar of the maps should be modified in order to see differences of the parameters from one region to another. Most points are green…
- The f**igure captions lack precision**. They do provide enough information and details on what the figure shows.

- **Summary:** in proposing a new coastal sea level product, this study would be of interest for the oceanographic and climate communities. However, in its current version, it lacks important basic information related to the data processing and to the comparisons with the ESA CCI product and the tide gauges. As a consequence, the results are not always convincing. I strongly recommend **major revision** and re-review of this manuscript and ask the authors to explain in details what they exactly did at the different steps of their analysis.

---

## Author Response (AR1)

**Reviewer(s)' Comments to Author: The IAS2024 coastal sea level dataset and first evaluations**

Thanks for very supportive and helpful comments. Our response to comments by reviewers is shown below in blue.

**Reviewer #1:**

In this paper, the authors present a new 20 Hz coastal altimetry product covering the worldwide coast from the Jason mission series during January 2002 to April 2022. The new product is generated by reprocessing the waveforms using the SCMR (Seamless Combination of Multiple Retrackers) strategy developed by the same authors and documented in published papers. By using this novel product, the authors build 1548 virtual stations, which can be used to monitor the coastal sea levels and calculate the vertical land motion (VLM) in combination with nearby tide gauges. A validation is made against tide gauges and a similar ESA product that uses a different improvement strategy (e.g., only one retracker). The new product is well documented with explanation of the processing that authors made, data analyses, comparisons with the independent data sets, and examples of exploitation.

Overall the paper has a clear structure. It is well written, detailed in data, methods and description of results. It is expected to become the first product shared though the International Altimetry Service. I feel the radar altimetry community and coastal oceanographers will appreciate the availability of this new high resolution global coastal altimetry data set, especially for long-term studies, although the product is still limited to the Jason series. Anyway, I am in favour that authors share their data with their peers who can suggest improvements or identify deficiencies. This will stimulate authors to extend the processing to the other missions.

In summary, the freely available new products has a clear potential for exploitation. The paper well explains it, therefore, I don't have serious concerns to suggest, except to better document validation results in the abstract and in the text. Having said that, I recommend minor revision.

Thanks for your confirmation, which encourages us to further improve the quality of the manuscript. In the revised manuscript, we have updated the comparison results with the ESA CCI product. Moreover, the CMEMS L3 along-track product is also used for comparison to further demonstrate the reliability of the IAS2024 dataset. The major changes in the revised manuscript include: 1) correcting the errors in interpolating the ESA CCI points into the nominal ground track segments; 2) the comparison between different altimeter datasets is conducted over the same ground track segments to better clarify the results; and 3) only the correlation coefficients, whose p-values are smaller than 0.05, are shown in the revised manuscript.

After doing this, the altimeter datasets show good consistency at common points in terms of high correlation coefficients (>0.4) and low root mean square values (40 mm-60 mm). The sea level trends from the IAS2024 dataset are on average  $1.32\pm2.40$  mm yr-1 higher than those from the ESA CCI v2.4 dataset, and are similar to those from CMEMS L3 product (-0.18±2.17 mm yr-1).

**Reviewer #2:**

This study proposes a new reprocessing of altimetry data from Jason-1, 2 and 3 in the world coastal zones over 2002-2022. It provides a new useful data set of altimetry-based sea level time series at a large number of coastal sites, complementing the limited tide gauge network. The coastal data are provided along satellite tracks at 20hz resolution, from the coast up to 20 km offshore. They are based on retracking of the radar waveforms using 3 different retrackers (ALES, WLS3, Adaptative Retracker), in addition to the standard retracker MLE4 used for the open ocean geophysical data records (GDRs). A total of 1548 'virtual' coastal stations are built along the world coastlines. The data set is validated by using available tide gauge data. It is also compared to a previously published, similar coastal product (called here ESA CCI product).

In principle, this proposed new dataset merits to be published. However, important clarification concerning the data processing is needed. The comparison with the ESA CCI product also requires significant clarification. Thus, the manuscript requires major revision before being suitable for publication in ESSD. Overall, the manuscript lacks a lot of crucial information about the data processing. Besides, the recurrent claim, all along the paper, that the IAS2024 is superior than the ESA CCI product (which is possibly true but not fully convincing) suggests competing interests rather than the desire of objectively presenting a new useful product.

Thanks for your useful comments, which help us to further improve the quality of the manuscript. In the revised manuscript, we have updated the comparison results with the ESA CCI product. Moreover, the CMEMS L3 along-track product is also used for comparison to further demonstrate the reliability of the IAS2024 dataset. The major changes in the revised manuscript include: 1) correcting the errors in interpolating the ESA CCI points into the nominal ground track segments; 2) the comparison between different altimeter datasets is conducted over the same ground track segments to better clarify the results; and 3) only the correlation coefficients, whose p-values are smaller than 0.05, are shown in the revised manuscript.

After doing these, the altimeter datasets show good consistency at common points in terms of high correlation coefficients (>0.4) and low root mean square values (40 mm-60 mm). The sea level trends from the IAS2024 dataset are on average  $1.32\pm2.40$  mm yr-1 higher than those from the ESA CCI v2.4 dataset, and are similar to those from CMEMS L3 product (-0.18±2.17 mm yr-1).

Moreover, we highlighted in the revised manuscript that our intention is to provide a new coastal sea level dataset for the ocean community instead of competing the ESA CCI product. The ESA CCI product is used for validating the reliability of the IAS2024 dataset.

Major and minor concerns are listed below (with major concerns in bold).

Page 6, Calculation of altimeter SSH: To my knowledge the GPD+ wet tropospheric correction is not available beyond mid-2021. Which correction is applied beyond that date?

The modelled wet tropospheric correction at zero altitude has been used instead of the GPD+ correction considering that the GPD+ is unavailable beyond mid-2021. It has been explained in the revised manuscript.

Page 6, Seamless combination of SSH estimates from multiple retrackers: it is not clear how the different retrackers are used. What is the bias (wrt the MLE4-based GDRs) you are talking about? Is it estimated for each 20Hz measurement? What is the Dijkstra algorithm? In fine, how are the ranges estimated? Are they based on different retrackers depending on the shape of the radar waveforms? Please clarify. I think it would be useful to provide some discussion on the respective

performances of the retrackers. The retracking procedure is crucial for building coastal altimetry data sets.

To better illustrate this issue, we updated Fig.1 in the revised manuscript. As we can see from the graph, the coastal retrackers (i.e., ALES, WLS3 and MB4) are applied to all 20-Hz waveforms to derived range estimates, respectively. Combined with satellite altitude, range and geophysical corrections, the SSH estimates for these coastal retrackers and SGDR MLE4 retracker are obtained. Considering the above retrackers can handle different types of waveforms as shown in Figure 3 and achieve similar performance through our previous Monte Carlo simulation (Peng and Deng, 2018, Peng et al., 2021), it is possible to further improve the data availability by combining SSH estimates from these retrackers. However, these retrackers adopt different processing strategies, there inevitably exist systematic biases between SSH estimates from these retrackers, which would prohibit the seamless combination of SSH estimates. To solve this problem, we proposed an interretracker bias model to reduce the SSH bias with respect to the SGDR MLE4 from centimeter levels to millimeter levels (Peng et al., 2024b). Finally, the optimal SSH estimate at each along-track point is selected with the Dijkstra algorithm. The Dijkstra algorithm is used because it can automatically determine the along-track SSH profile with smooth spatial variation (Roscher et al., 2017), which is consistent with the real situation that the altimeter along-track SSH estimates do not change significantly at the spatial scale of ~300 m (Cipollini et al., 2017). Moreover, the Dijkstra algorithm does not rely on the waveform classification results. This has been illustrated in the revised manuscript.

Page 7, Assessment of the IAS2024 dataset : After your 3-sigma editing, how are the temporal distribution of the data per mission? For example, the Jason-1 mission suffers more for missing data than Jason-2 and Jason-3. These missing data affect trend estimates. Please discuss.

To remove the SLA outliers, the 3-sigma filter is applied twice. First, it is applied to the along-track SLA estimates. Second, it is applied to the SLA time series. To guarantee the robustness of the trend estimates, only the SLA time series whose percentages of data availability are higher than 80% are used for the estimation. This is consistent with the strategy adopted by ESA CCI product (Birol et al., 2021). Therefore, the impact of missing data on trend estimation can be ignored. This has been explained in the revised manuscript.

Page 8, Assessment of the IAS2024 dataset: Little is said about the intermission bias estimation. Since you are focusing on coastal altimetry data, how do you compute it? Locally? regionally? This point is also crucial because any error in the intermission bias directly affects SLA trend estimates.

We have compared the performance of different methods used to estimate the inter-mission bias for Jason missions in our previous study (Peng et al., 2022). The results show the overlapping time series is most suitable for the bias estimation. The bias is estimated as the mean difference between time series from two altimetry missions within the overlapping period for each 20-Hz along-track point. This has been explained in the revised manuscript.

Page 8 and beyond, Assessment of the IAS2024 dataset : How is the coast defined? Which data set is used and how is the distance to the coast computed?

The offshore distance is derived from the Jason-3 SGDR product. In this study, the Jason-3 ground tracks which achieve the maximum data points within 100 km to the coast are selected as the nominal ground track points. This has been explained in the revised manuscript.

Do you apply a smoothing to your SLA time series?

**We do not smooth the SLA time series.**

Could you show the distribution of closest distance to the coast for the 0-10km segments?

**It has been shown as Fig. 4 in the revised manuscript.**

Page 9 and beyond , Application of the IAS2024 data set: You define three sets of along-track segments (0-10km, 5-15 km, 10k-20 km) and assign virtual stations to each segment. From Table 4 (page 16), I note about the same number of virtual stations are found for each segment, which is surprising. I would expect more virtual stations for the offshore segment (10-20 km) than for the onshore segment (0-10km) because more noisy data should have been eliminated in the latter case. Please discuss.

As we can see from Table 5, the number of virtual stations (1359) over 0-10 km is smaller than that (1548) over 5-15 km and 10-20 km. This is because the data quality is significantly degraded within 5 km to the coast. This has been illustrated in the revised manuscript.

There is no indication on how the 10-day data are projected onto fixed points along a nominal ground track. How do you proceed?

The 20-Hz along-track SLA estimates from 10-day repeat cycles are referenced to the corresponding ground track segments using the nearest neighborhood approach. This has been illustrated Section 2.2 in the revised manuscript.

Page 10, Figure 4: indicate on the panels the name of the satellite. The figure caption should be more detailed.

**This has been revised.**

Page 11-14: Comparison with the ESA CCI data set: I have a number of comments on this comparison.

First of all, it is not clear at all which version of the ESA CCI product has been used here. Page 8, it is written that the ESA CCI v2.3 data set is used (with data provided over January 2002 to June 2021). If this is correct, why only ~800 sites are correlated in Figure 6.a (I exclude the left hand part of the histogram centered at zero which seems to correlate data from IAS2024 at sites where ESA CCI has no data; i.e., in high latitude regions, Red Sea, tropical islands –see my comment below-). The ESA CCI v2.3 version contains 1189 virtual stations. Thus the histogram should show 1189 counts. The authors should indicate the date at which they downloaded the ESA CCI data to be clear about which product they considered.

Thanks for your comments. We found that the ESA CCI dataset has been updated to version 2.4. Therefore, we download the latest version for comparison and update the corresponding information in the revised manuscript as "European Space Agency (ESA) Climate Change Initiative (CCI) coastal sea level dataset (version 2.4, https://www.seanoe.org/data/00631/74354/, last accessed at December 5, 2024)."

We also found that there contain 1132 virtual stations in the ESA CCI v2.4 dataset. However, when we interpolated the ESA CCI data into the ground track segments used in this study, only 807 ground track segments have at least 10 common points with the IAS2024. This is because the ESA CCI product has no data in places such as the northern Europe and Japan. In contrast, the number for

**IAS2024 dataset is 1458.**

To avoid impact of different spatial coverage of these two datasets, the comparison is conducted over the same ground track segments. Moreover, when calculating the correlation coefficient, we only keep the results whose p-values are smaller than 0.05 and now the results are similar to those provided by the reviewer 4. The updated results are shown in Figures 7 and 8 in the revised manuscript.

It is not fully clear how the ESA CCI data are used for the comparison. From what is written page 12, lines 261-264, it looks like that the ESA CCI along-track SLAs are averaged over the same 3 segments as for the IAS2024 data and then compared. But according to Fig.6a 'correlation coefficient plots' this seems not be the case. How many common points are considered in each segment? Does this comparison refer to the common time span of the two data sets? 6a is totally unclear and likely misleading. To which virtual stations correspond the two histograms? As the processing of the raw coastal altimetry data are rather similar (except that IAS2024 uses several retrackers compared to ESA CCI which only uses ALES, but all other corrections being the same), I do not understand the large number of cases with zero correlation. This seems totally unlikely. My understanding is that the left hand side histogram of Fig.6a centered at zero corresponds to the purples dots of Figure 7 (i.e., where the ESA CCI product has no data). If this is the case, then the histogram centered at zero is not only misleading but definitely wrong! Please remove it!

This is due to the interpolation errors as we mentioned before. After correcting this error, the IAS2024 and ESA CCI data are compared over the same 807 ground track segments where at least 10 common points are available over the last 20 km to the coast for these two datasets. The time spans for these two datasets are very close and would not affect the comparison of sea level trends. In addition, we only keep the correlation coefficients whose p-values are smaller than 0.05. To make it more clearly, it has been illustrated in the revised manuscript. The results in Figures 7 and 8 have now been updated following the above strategy. It is found that the correlation coefficients mostly vary between 0.4 and 0.8, indicating the good consistency between these two datasets.

Please show examples to SLA time series at the same location from the two products. If, for example, your SLA time series are smoothed and if you compared to unsmoothed ESA CCI data, the correlation will indeed decrease. Please explain exactly how you have done.

7a is misleading. As the ESA CCI product has no data in high latitude regions, red sea and tropical islands, the map should show the correlation ONLY at the common sites. All purple points (where ESA CCI has no data) need to be removed to not give the false impression that there are numerous sites where the correlation is zero.

7b: same comment as for Fig.7a. It is misleading to show high rms (red points) where the ESA CCI product has no data!

We do not use the smoothed data for comparison. Thanks to the reviewer 4, the examples of SLA time series from both IAS2024 and ESA CCI v2.4 are shown. The zero correlation coefficients are due to the errors in the interpolation of the ESA CCI points into the nominal ground track segments where ESA CCI has no data. The Figure 8 is also updated in the revised manuscript.

Page 13: Comparison with ESA CCI; trend differences. As for the SLA time series, it is totally unclear how the comparison is done. To trust this comparison exercise, you must be clear on what you do.

As mentioned in the revised manuscript, the comparison of sea level trends is similar to that of correlation coefficients. First, the point-wise sea level trends and trend uncertainties are derived for

both IAS2024 and ESA CCI v2.4 datasets. Then, the trend differences and uncertainties at common points are averaged for each ground track segment. Finally, the histogram of both trend differences and uncertainties are shown in Figure 8. This has been illustrated in the revised manuscript.

I recommend you provide a comparison of your trends computed over the 10-20km segments with trends estimated from classical gridded products (e.g., Copernicus datasets, CMEMS or C3S). This would be another highly useful way of validating your product (in terms of trends). This has been done for the ESA CCI product for the offshore data (>10km from the coast) with trend differences < 1-2 mm/yr in general with the C3S gridded product.

Thanks for your suggestions. Because the gridded products merge altimeter data from multiple altimetry missions, we use the CMEMS Level-3 along-track product from Jason missions to validate our product. The updated results are shown in Section 3.3 in the revised manuscript, and the good consistency between these two datasets is observed.

Page 14-16, Altmeter-based virtual stations: comparison with tide gauges. How many tide gauge are used? Justify the 200 km distance threshold considered for the virtual stations-tide gauge comparison. Do the tide gauge records have continuous data over the overlapping time span or not? If not, how do you proceed? Please explain.

549 PSMSL tide gauges are selected, which have continuous data records over the period of 2002 and 2022. The 200 km distance threshold is selected due to the trade-off between including enough nearby virtual stations and retaining high correlation coefficients. This has been illustrated in the revised manuscript.

Table 4 as well as Fig.14d and page 21: Concerning the trend comparison, what means 'trend differences'? Do the authors compare VLMs estimated from the differences altimetry-based SLAs and tide gauges, and GNSS-based VLMS? If this is the case, how many tide gauges have collocated GNSS stations? In many cases, GNSS-based VLM solutions often show significant discrepancies between computing centers. You chose the ULR7a solution here, which is fine. But do you have an idea of the associated VLM uncertainties?

Yes. The vertical land motion is calculated from the differences of de-seasoned monthly SLA time series between altimeter and tide gauge. The ULR7a contains 546 stations. In this study, the PSMSL tide gauges and ULR7a stations are collocated if their distance is within 10 km, and the number of collocated stations is 166. The VLM uncertainties from ULR7a are mostly within the range of 0 mm yr-1 and 1 mm yr-1, while the corresponding uncertainties from the combination of tide gauge and altimeter vary between 0.5 mm yr-1 and 2 mm yr-1 (Fig. 20). This has been illustrated in the revised manuscript.

Pages 20-21, Vertical land motions: This section estimates VLMs from the differences altimetrybased SLAs and tide gauge data. This is ok. But over the rather short time span of analysis (20 years), many coastal SLAs display strong interannual variability, masking any long term trend. Only a few coastal regions show dominant linear sea level trends over this time span. Thus what are the errors associated with the trend estimates by this approach?

The major errors come from different spatial and temporal sampling manners between altimeter and tide gauge, different geographical locations between altimeter-based virtual stations and tide gauges as well as the short time span of the time series. This has been illustrated in the revised manuscript. The plots in Figures 15d and 16b are unclear. What are supposed to show the boxes? The color bar of the maps should be modified in order to see differences of the parameters from one region to another. Most points are green...

The figure captions lack precision. They do provide enough information and details on what the figure shows.

The boxplot shows the trend differences between altimeter-derived and GNSS-derived VLM. On each box, the central mark indicates the median, and the bottom and top edges of the box indicate the 25th and 75th percentiles, respectively. The whiskers extend to the most extreme data points are not considered outliers, and the outliers are plotted individually using the '+' marker symbol. The color bar has been modified in the revised manuscript. Moreover, the figure captions are adjusted to give more information.

Summary: in proposing a new coastal sea level product, this study would be of interest for the oceanographic and climate communities. However, in its current version, it lacks important basic information related to the data processing and to the comparisons with the ESA CCI product and the tide gauges. As a consequence, the results are not always convincing. I strongly recommend major revision and re-review of this manuscript and ask the authors to explain in details what they exactly did at the different steps of their analysis.

Thanks for your valuable suggestions. In revised manuscript, we correct the errors in interpolating of ESA CCI data into the nominal ground track segments and update the comparison results between IAS2024 and ESA CCI. Moreover, the CMEMS L3 product is also used for comparison.

**Reviewer #3:**

This new altimetry dataset is important for studying coastal sea level variability on a global scale and I appreciate this work. But I do, however, have a few comments on this study.

My main concern is the comparison with the CCI coastal sea level product which is an important component of this paper. It's a really good thing that at least 2 similar products exist and can be compared. This always allows to improve the calculation methods and ultimately the data quality. But I don't understand the results in Figure 7 (and therefore Figure 6). To my knowledge, the ESA CCI v2.3 dataset has actually less segments than IAS2024 (as written in the paper), does not cover all the high latitudes and covers almost no island (i.e. most of the purple and red points in figure 7). So I don't understand how the correlation and RMS statistics were calculated between the IAS2024 and ESA CCI v2.3 time series at these points where there is no CCI data? This could explain the 2-lobe distribution in Figure 6, but not all the values are 0 either. So can the authors give more details on the method of comparison between the 2 datasets? Is there a spatial interpolation between the IAS24 point and the nearest CCI point? And is there a distance limit between the points in the 2 datasets for calculating the statistics?

Thank you very much for arising this problem. We carefully checked the MATLAB codes and found that there is an interpolation error. Therefore, we have corrected the errors in interpolating the ESA CCI data into the nominal ground track segments and updated the comparison results between IAS2024 and ESACCI in Section 3.2.

In other words, in the conclusion it is written: "The spatial distribution of both correlation and coefficients and RMS values (Fig. 7) reveals that the discrepancy between IAS2024 and ESA CCI v2.3 datasets is mainly observed in the mid-to-high latitudes (>40°N and <40°S) and islands over open oceans". Is this not just due to the fact that there is no CCI data for most of these points?

**You are correct. The ESA CCI has no data in these places. After correcting the interpolation errors, the good consistency between IAS2024 and ESA CCI is clearly presented in the revised manuscript.**

Similarly, it is really important to analyse whether or not the 2 products give the same information in terms of coastal sea level trends. Especially because of the complexity of coastal altimetry processing. This adds confidence in the maturity of current data processing. To verify this, it would be interesting to add to Table 3 the IAS product statistics calculated on exactly the same stations as the CCI product. The same applies to Figure 12 (if this is not the case).

Thank you for valuable suggestions. In the revised manuscript, the same ground track segments for both IAS2024 and ESA CCI have been used for comparison to make the results more clearly. Moreover, the CMEMS L3 1-Hz along-track product has also been used for comparison.

Best regards,

**Florence Birol**

Reviewer #4:

To better understand the comparison made between the CCI and the IAS24 coastal virtual stations we choose to perform the same comparison. To achieve this we choose to compare only data where both datasets have common along-track point. We additionally conduct comparisons with tide gauge data obtained from PSMSL. This led us to different results that we want to share with you. You can find the report on the following link:

https://filesender.renater.fr/?s=download&token=5d4e9da1-dd60-40e4-b7a8-8b9bb6e3e71e

Best regards,

Lancelot Leclercq, member of ESA CCI sea level project

Thank you for providing the comparison results, which help us to find out where the problem is. In the revised manuscript, we have updated the comparison results with the ESA CCI product. Moreover, the CMEMS L3 along-track product is also used for comparison to further demonstrate the reliability of the IAS2024 dataset. The major changes in the revised manuscript include: 1) correcting the errors in interpolating the ESA CCI points into the nominal ground track segments; 2) the comparison between different altimeter datasets is conducted over the same ground track segments to better clarify the results; 3) only the correlation coefficients, whose p-values are smaller than 0.05, are used in the revised manuscript.

After doing these, the altimeter datasets show good consistency at common points in terms of high correlation coefficients (>0.4) and low root mean square values (40 mm-60 mm). The sea level trends from the IAS2024 dataset are on average  $1.32\pm2.40$  mm yr-1 higher than those from the ESA CCI v2.4 dataset, and are similar to those from CMEMS L3 product (-0.18±2.17 mm yr-1).

---

## Referee Report (RR1)

Report on 'The IAS2024 coastal sea level dataset and first evaluations' by Peng et al. -Revised version-

I (Reviewer 2) have read the revised manuscript and found it considerably improved compared to the original version. The authors have well responded to all my comments. This revised version now reads very well. It includes the detailed information on the data processing that was lacking in the initial version. It has also improved the comparison with other data sets. I think this revised version is now acceptable for publication in ESSD.

I list below a few minor comments that should be taken into account by the authors when preparing the final version.

- Abstract, line 2: change 'has been' by 'is'
- Lines 40-41: change '…calculate…estimates' by '…estimate the vertical land motions.'
- Line 136: explain what is the Topex ellipsoid and give a reference
- Table 1: give references for EWCMF and CLS22
- Line 144: change 'derived' in 'derive'
- Line 161: define SWH
- Line 172: give a reference for the CLS22 mean sea surface
- Line 181: add a sentence, explaining that crossovers can occur between ascending and descending Jason tracks
- Line 186: add the reference to Holgate et al., 2013 for the PSMSL
- Line 187: Figure 13a showing the location of the 549 tide gauges should appear here or at least at the beginning of the paragraph that starts line 330 or line 345
- Line 248: give a reference for ITRF 2014
- Figure 9: explain in the figure caption what the violet bars represent
- Line 294: change 'They' by 'the two datasets'
- Lines 337-338: cancellation of global VLMs may be true on a 'true' global scale. But the distribution of the 549 tide gauges used in this study is far from global. Thus I am not sure that cancellation applies here
- Line 356: change 'where' by 'which'
- Lines 505-506: differences in altimetry-based sea level trends and tide gauge trends are visible in several other regions

A final comment: I do not see in the text any information on how the coast is defined and on the associated dataset. Please correct this.

---

## Author Response (AR2)

**Reviewer(s)' Comments to Author: The IAS2024 coastal sea level dataset and first evaluations**

Thanks for very supportive and helpful comments. Our response to comments by reviewers is shown below in blue.

Reviewer #1:
I (Reviewer 2) have read the revised manuscript and found it considerably improved compared to the original version. The authors have well responded to all my comments. This revised version now reads very well. It includes the detailed information on the data processing that was lacking in the initial version. It has also improved the comparison with other data sets. I think this revised version is now acceptable for publication in ESSD.

I list below a few minor comments that should be taken into account by the authors when preparing the final version.

Thank for you confirmation and the manuscript has been revised as you suggested.

- Abstract, line 2: change 'has been' by 'is'

It has been revised.

- Lines 40-41: change '...calculate...estimates' by '...estimate the vertical land motions.'

It has been revised.

- Line 136: explain what is the Topex ellipsoid and give a reference

The Topex ellipsoid is a reference ellipsoid which has an equatorial radius of 6378.1363 kilometers and a flattening coefficient of 1/298.257.

- Table 1: give references for EWCMF and CLS22

It has been added.

- Line 144: change 'derived' in 'derive'

It has been corrected.

- Line 161: define SWH

It has been revised.

- Line 172: give a reference for the CLS22 mean sea surface

The reference has been added.

- Line 181: add a sentence, explaining that crossovers can occur between ascending and descending Jason tracks

The sentence has been revised as "the crossover point is the intersection between the ascending

and descending ground tracks".

- Line 186: add the reference to Holgate et al., 2013 for the PSMSL

The reference has been added.

- Line 187: Figure 13a showing the location of the 549 tide gauges should appear here or at least at the beginning of the paragraph that starts line 330 or line 345

The Figure 13 has been changed into Figure 4 in the revised manuscript.

- Line 248: give a reference for ITRF 2014

The reference has been added.

- Figure 9: explain in the figure caption what the violet bars represent

The violet bars are the overlaps between IAS2024 and ESA CCI. It has been mentioned in the revised manuscript, as well as the Figure 13.

- Line 294: change 'They' by 'the two datasets'

It has been revised.

- Lines 337-338: cancellation of global VLMs may be true on a 'true' global scale. But the distribution of the 549 tide gauges used in this study is far from global. Thus I am not sure that cancellation applies here

As illustrated by Wöppelmann and Marcos (2016), a possible average bias would be obtained when the number of sites decreases. This means the cancellation of global VLMs can still be achieved at regional scale, which is the case in the study. To clarify this, we revised the sentence as "The result from the IAS2024 dataset is thus consistent with this assumption even it is not a true global scale, as the mean of trend differences is –0.26±3.57 mm yr$^{-1}$".

- Line 356: change 'where' by 'which'

It has been revised.

- Lines 505-506: differences in altimetry-based sea level trends and tide gauge trends are visible in several other regions

We agree and here we only denote the remarkable difference between tide gauge and altimeter in the revised manuscript.

A final comment: I do not see in the text any information on how the coast is defined and on the associated dataset. Please correct this.

The world's coastline is defined by the Global Self-consistent, Hierarchical, High-resolution Geography Database (GSHHG) dataset (https://www.soest.hwaii.edu/-pwessel/gshhg/), which is used by the official SGDR product to calculate the offshore distance, which is explained in Section 2.2 in the revised manuscript.